# A survey on text classification: Practical perspectives on the Italian language

Andrea Gasparetto[1]*, Alessandro Zangari[1], Matteo Marcuzzo[1], Andrea Albarelli[2]

**1** Department of Management, Ca' Foscari University, Venice, Italy, **2** Department of Environmental Sciences, Informatics and Statistics, Ca' Foscari University, Venice, Italy

☯ These authors contributed equally to this work.
* andrea.gasparetto@unive.it

**Data Availability Statement:** All data and code used for the experimental session have been linked in the article, except for Reuters Corpus Volume 1 and 2, which, as a dataset owned by a third party, cannot be shared, but must be requested to the

## Abstract

Text Classification methods have been improving at an unparalleled speed in the last decade thanks to the success brought about by deep learning. Historically, state-of-the-art approaches have been developed for and benchmarked against English datasets, while other languages have had to catch up and deal with inevitable linguistic challenges. This paper offers a survey with practical and linguistic connotations, showcasing the complications and challenges tied to the application of modern Text Classification algorithms to languages other than English. We engage this subject from the perspective of the Italian language, and we discuss in detail issues related to the scarcity of task-specific datasets, as well as the issues posed by the computational expensiveness of modern approaches. We substantiate this by providing an extensively researched list of available datasets in Italian, comparing it with a similarly sought list for French, which we use for comparison. In order to simulate a real-world practical scenario, we apply a number of representative methods to custom-tailored multilabel classification datasets in Italian, French, and English. We conclude by discussing results, future challenges, and research directions from a linguistically inclusive perspective.

## Introduction

Text Classification (TC) is one of the most essential tasks in the field of Natural Language Processing (NLP). This denomination is usually associated with a broad category of more specific procedures, which roughly share the common objective of designating predefined labels for a given input body of text. Over the years, TC procedures have evolved from simple, rule-based systems to highly specialized architectures. The latter have gone closer than ever before to showing actual understanding of the underlying semantics of a piece of text, utilizing such meaning in order to make an informed decision for the classification process.

There are countless practical applications of TC, including information retrieval, topic labeling, sentiment analysis, and news classification. Even more loosely related tasks, such as extractive text summarization and content-based recommendation systems, can be approached within a TC framework.

National Institute of Standards and Technology (NIST). The Reuters Corpus Volume 1/2 is a collection of Reuters News stories that can be used for research purposes. Volume 1 contains only English articles, while Volume 2 contains articles in 13 different languages. Articles are annotated with various hierarchical labels, including topic codes which have been used as a general descriptor of the article's content. All information on how to request this data are described on the following website: https://trec.nist.gov/data/reuters/reuters. html. Once data has been obtained, the procedure to generate the dataset and reproduce the presented experiments is described in the code repository available here: https://gitlab.com/ distration/dsi-nlp-publib#rcv1-rcv2-news-categorization.

**Funding:** The author(s) received no specific funding for this work.

**Competing interests:** The authors have declared that no competing interests exist.

Due to the speed at which textual information is produced, it has become essential to rely on automatic processing techniques to handle continuously increasing volumes of data. However, the adoption of modern machine learning (ML) methods in this context can be non-trivial. Recent ML methods rely on the ingestion of massive amounts of textual documents in order to effectively model a probability distribution over sequences of words. Hence, the limited availability of text corpora (i.e., large collections of digitized textual data) in some languages constitutes a serious obstacle to the application of these methods. Such resources are essential to the development of modern approaches and add to the intrinsic difficulties of this task.

## Resource categorization of language

In order to better understand how the resources tied to a language influence the application of ML algorithms, we briefly discuss the topic of resource categorization of languages. In the field of computational linguistics, it is common to define language as low-, mid- or high-resource. While there is no standardized approach to determine whether a language fits into one category or the other, a reasonable categorization is usually easy to find and agree on. The resources described by these denominations refer to raw data (i.e., collections of digitized text) as well as linguistic tools and software necessary to perform various tasks. In the particular context of Text Classification, tools like these might be needed to perform common text interpretation procedures such as lemmatization and part-of-speech (PoS) tagging (further outlined in the Preprocessing section).

As there is no standardized approach to this classification, the spectrum of resources in which languages lie is highly speculative, but it is fair to claim that the most well-resourced end is dominated by English and Chinese (Mandarin). Other languages commonly considered as high resource include Arabic, French, German, Portuguese, Spanish and Finnish, though most of research implicitly utilizes one of the former two languages, English in particular [1].

Throughout this survey, we utilize Italian as a means of comparison. As a language, Italian can be considered on the higher end of this spectrum, somewhere between a mid- and high-resource categorization. Indeed, as far as raw computerized text data is concerned, it is a rather well documented language. However, task-specific data, indispensable to test and validate TC algorithms, can be severely lacking for this language. Moreover, sets of textual data for specific downstream tasks may only be available if duly licensed (and sometimes at a cost), something that is certainly true for Italian as well as many other languages. Obviously, this can vastly limit the potential for research.

## The importance of generalizing linguistic research

Much of modern research focuses on a few, dominant high-resource languages like English. However, something that is certainly desirable for any NLP model is for it to be validated on its capability of generalizing its result on data and languages other than those on which it has been trained and tested [2]. This concept is relatable to that of *language independence* [3], an attribute that describes models that can be made to work comparably well across languages.

Because this aspect is often overlooked, applying Text Classification procedures can be challenging in languages other than English. This might be because linguistic tools are lacking (or simply perform differently), or because suitable benchmarks are not readily available. Furthermore, as research has moved more and more toward deep learning methods, the divide can be further exacerbated by the interpretability issues of these models, which makes it difficult to ascertain their effectiveness on other languages.

## Major differences and contributions

In recent years, multiple excellent works have reviewed TC from a generic, language-agnostic perspective. Li et al. [4] provide a comprehensive investigation of models ranging from traditional approaches to more deep learning-based models. We follow their excellent categorization of approaches. Kowsari et al.'s [5] survey is notable for its in-depth exploration of stages such as feature extraction and dimensionality reduction, which are more common in traditional approaches. Minaee et al.'s [6] work focuses solely on a thorough exploration of deep approaches, though it notably also provides quantitative results for classical methods in its experimental performance analysis. The main objective of this work is to provide insight into the main linguistic challenges involved in the development of TC methods as applied to languages other than English. While we provide a brief summary of some of the most prominent TC techniques, we emphasize those aspects related to the linguistic component of this task.

The main language studied while surveying these methods was Italian; to reiterate, this is a well-documented language, for which we will however showcase a scarceness of task-specific datasets. To this end, we provide a list of notable TC datasets for Italian and complement it with a similarly built list for the French language, such as to provide a fair comparison with a high-resource language that is not necessarily English. We describe how to distill a multilabel dataset for topic labeling from Wikipedia dumps, as well as a news classification dataset from Reuters articles. We perform a study of compatibility between a set of representative algorithms and these two datasets for the multilabel classification task, for which we discuss various challenges and difficulties encountered. In summary, this study's main contributions are as follows:

- We provide a high-level overview of TC, highlighting which steps of the pipeline have been shown to be more language-dependent;

- We highlight recent developments in classification methods for NLP, including modern preprocessing operations and pre-trained language models;

- While introducing the operation of a TC pipeline, we discuss the main causes of compatibility issues with languages other than English;

- We demonstrate the applicability of several traditional and modern methods to multilabel datasets in three different languages (Italian, French, and English);

- We underline technical challenges and the current research directions being explored to solve them.

The rest of this survey is organized as follows. The first section discusses text preprocessing, going into detail about language segmentation as the most relevant operation from a linguistic point of view. We then discuss text representation techniques utilized to project preprocessed text into a feature space, briefly describing early methods and how they evolved into contextualized and semantically meaningful vectorial embeddings. We discuss the issues posed by the computational expensiveness of these methods, and why these are problematic for their application in multiple languages. We dedicate a short section to classification algorithms and how their importance has diminished in favor of better text representation. The latter sections deal with experimental factors, describing TC tasks and showcasing datasets in Italian and French, outlining the search criteria, and providing a selection of English datasets for comparison purposes. We provide quantitative results for a select choice of multilabel datasets in all three languages. Finally, we summarize the main future challenges faced by TC methods, before

concluding the survey. Datasets and code used for the experimental part of this work are available (when legally possible) at https://gitlab.com/distration/dsi-nlp-publib.

## Preprocessing

A fundamental part of the Text Classification pipeline resides in its preprocessing steps. Raw textual information is *unstructured* and does not have a straightforward numerical representation (differently, for example, from types of data such as images). Clearly, from a linguistic point of view, languages are indeed ruled by a very complex structure, one that might be intuitive to a native speaker of that language, but much less so to a machine.

It becomes therefore necessary to project text into an appropriate feature space so that it can be handled by a learning algorithm. In this section, we discuss all those procedures that prepare textual data for this projection, whether this is done through manual feature extraction (as with earlier, more traditional methods) or automatically (as is with recent, deep learning-based approaches). We provide an overview of the most important preprocessing operations, while the section that follows will describe possible choices for obtaining machine-friendly representations from the resulting preprocessed text. We place particular focus on tokenization as, among the early steps of language interpretation, it is certainly the most critical, having a considerable impact on downstream performance on several NLP tasks [7].

### Tokenization

The first and most basic operation is that of *tokenization*, the process of breaking a stream of text into smaller chunks (historically called *tokens*). The most traditional as well as intuitive atomic unit of choice (i.e., token) has been centered around words [8]. Recent approaches have instead been applying more granular decomposition processes, such as character *n*-grams, sub-words and, most recently, even segmentation approaches based on the underlying byte representation of text [9]. It has been argued that, among preprocessing operations for any NLP task, tokenization can be regarded as the most important language-dependent operation [10].

The following sections will describe the main difference between more traditional and recent tokenizers, showcasing an interesting trend towards maximal decomposition. It is worth mentioning that, as of now, researchers agree that there is no single best solution, and the choice of unit of text is one that must be made depending on the context and necessity of the application.

**Pre-tokenizers.**   Conventional approaches to the tokenization task have traditionally been *rule-based*, and, especially in most white-spaced writing systems (i.e., languages where spaces are used as word separators in writing), minimal tokenization can be carried out by separating around blank characters, punctuation, and contractions. Clearly, this intuitive approach has seen many refinements, often integrating language-specific knowledge into its rules. While not perfect, such segmentation approaches are deemed an acceptable approximation of actual morphemes, striking a compromise between linguistic irrelevance and purely typographic tokens [8]. Examples of popular rule-based tokenizers include Moses [11], and the SpaCy tokenizer [12]. Both Moses and SpaCy are NLP toolkits and include tokenizers that work with multiple languages using a set of language-specific rules and exceptions.

Recent literature often defines earlier tokenization approaches as "pre-tokenizers", because of how many modern methods may use them as an initial step (therefore preceding "proper" tokenization).

**Data-driven tokenizers.** Tokenization (and language segmentation in general) has evolved greatly in recent decades. Here, we introduce some of the latest developments in the field, such as to highlight their close relationship with language representation approaches.

When provided with textual data, a tokenizer will decompose it and create a "vocabulary" of terms. At a practical level, this vocabulary is used to generate an index-based mapping between actual tokens and a numerical representation (different depending on the feature extraction technique). Modern text representation techniques are based on *embeddings*, rich vectorial representations which we will cover in the Text representation section. As each token in the vocabulary corresponds to a possibly large embedding, these representations are unable to handle arbitrarily vocabularies of arbitrary size because of time and space limitations. As a consequence, most modern language representation techniques require a fixed-size vocabulary.

It is clear, then, that modern tokenization approaches must strike a balance between the expressiveness of the vocabulary and its dimension. This expressiveness is most closely tied to the concept of out-of-vocabulary (OOV) words, which correspond to text units that have not been seen during a model's training. As such, the model is unable to extract useful information from OOV tokens (models such as these are termed as *closed-vocabulary*) [8]. A sufficiently expressive vocabulary, then, should be able to minimize the number of OOV terms, such as to fully utilize the information available at inference time.

OOV words are a central weakness of traditional tokenization approaches. Because of phenomena such as derivations, inflections, and contractions, certain languages can be difficult to segment properly, creating excessively large vocabularies. A solution can be to reduce tokenization to a character-level segmentation; while this has been tested with some degree of success, in many languages it can be hard to obtain a meaningful representation for single characters since they appear in too many different contexts and are not as relevant as, for example, words in terms of sequence modeling [13]. Furthermore, since each character is mapped to its own vector of parameters, the memory footprint increases for longer sequences. Many modern neural language representation approaches resort to truncation of input sequences to a pre-defined length in order to handle memory issues; doing this with character tokenization would mean keeping the first $k$ characters instead of the first $k$ words, potentially losing much of the original sequence information.

As both of these simple strategies are not entirely satisfactory, modern tokenization approaches most commonly employ hybrid techniques that split text into *sub-words*. Notably, while manually constructed approaches to this type of segmentation have been tested, the more popular method of choice for recent methods relies on automatically learning morphological segmentation in an unsupervised manner. The general idea of data-driven tokenizers is that frequently used words should not be split into smaller words, while rare words should be broken into more "reusable" fragments; this way, OOV tokens can be recognized as a composition of multiple known sub-words. In the following paragraphs, we introduce some of the most popular tokenizers that have seen widespread adoption in modern NLP models in recent years. Table 1 provides a concise view of the main modern tokenizers.

*Byte Pair Encoding.* An important breakthrough in tokenization strategies was the development of Byte Pair Encoding (BPE) [14], originally proposed as a data compression algorithm [19] and later adapted for sub-word segmentation. After a character-level pre-tokenization, smarter tokenization is learned by iteratively computing the co-occurrence of consecutive pairs of vocabulary terms, and merging the most frequent into a new vocabulary word. The same process is then applied when tokenizing unseen documents, executing recorded merges in the same order as they were during training. A notable extension of this segmentation

**Table 1. Most widely adopted recent tokenization approaches.**

| Tokenizer | Training Procedure | Inference Procedure | Language Support |
|---|---|---|---|
| BPE [14] | Merge most frequent consecutive pairs of *n*-grams | Merge incrementally, keeping merged term if in vocabulary | White-spaced only |
| BBPE [15] | Same as BPE, based on bytes instead of *n*-grams | Same as BPE | All languages |
| WordPiece [16] | Merge sub-words that maximize LM likelihood | Find longest first substring of words within vocabulary | White-spaced only |
| UnigramLM [17] | Start from pre-generated vocabulary, remove sub-words that least contribute to the LM likelihood function | Substring likelihood maximization through Viterbi Algorithm | All languages |
| SentencePiece (sw package) [18] | Fast, optimized procedures for other algorithms | Enhanced inference methods | All languages |

procedure is byte-level BPE [15], which applies the same algorithm not to characters but to raw bytes.

*WordPiece*. The WordPiece tokenizer [16] was initially developed for Japanese text segmentation problems, and relies on the creation of *n*-gram-based language models (in the classical sense, as we describe later in the Text representation section) to recognize recurring syllables, prefixes and word segments in a corpus. A greedy process iteratively increases the vocabulary size, starting from single characters, selecting and merging pairs of sub-words that maximize the language model likelihood. The algorithm stops when the expected likelihood falls below a predefined threshold, or the maximum vocabulary size is reached.

*UnigramLM*. Conceptually similar to WordPiece, UnigramLM [17] proceeds in the opposite direction, starting from a large vocabulary obtained by pre-tokenization and iteratively removing the terms with the lowest expected probability with regards to a simple unigram language model. The process is repeated until the desired size is reached. Multiple segmentations are possible due to the stochastic nature of this process, and while the most likely segmentation is chosen in practice, it is possible to implement sampling procedures to perform what is defined as "sub-word regularization", which has empirically been shown to improve results on some tasks.

*SentencePiece*. SentencePiece [18] is not an algorithm in itself but rather a software package containing optimized versions of the above approaches. Among other segmentation optimizations, it is a particularly worthy mention as it addresses the fact that other tokenizers depend on knowing which characters act as word separators in the corpus, which is language-dependent and may require specific pre-tokenization procedures to create rules to recognize word boundaries. Instead, SentencePiece considers text as a raw stream of characters, including word-separators, removing this operational constraint.

**Linguistic aspect of tokenization.** The segmentation of textual data into sentences and words has been historically rooted in linguistic motivations (as well as technical constraints). The common and intuitive approach of segmenting into words has the advantage that, from a linguistic point of view, these units can be labeled with linguistic annotations such as PoS tags (e.g., noun, verb) and syntactic dependency information (related to the structure of sentences) [8]. Therefore, utilizing linguistically motivated units opens the possibility of using such additional information throughout the classification pipeline.

However, it is not trivial to define and identify linguistic units, most notably because of the vast number of irregularities and language-specific phenomena involved. Works such as the Morpho-Syntactic Annotation Framework (MAF) ISO standard [20, 21] identify linguistic units as *word-forms*: these are represented by a stem and a list of inflections to be attached. For example, many English words can be inflected as verbs, adverbs, nouns, and adjectives. Word-

forms cover many linguistic phenomena, such as contractions (e.g., "*isn't*"), compounds (e.g., "*football*"), morphological derivatives (e.g., "*sadness*"), diminutive or augmentative derivations and more. Nevertheless, deriving a precise procedure to segment into word-forms is hard and expensive, and word-based segmentation is usually accepted as a reasonable approximation.

Similarly, other works focus on morpheme-based tokenization for morphologically complex languages [22–24]. Morphemes are the indivisible basic units of language that carry semantic meaning; learning meaningful context-independent representations of morphemes is challenging, particularly for agglutinative languages, where words can be composed by (almost) arbitrarily long and complex sequences of morphemes with minimal contextual change. This is in contrast to fusional languages, where morphemes are stitched together usually with more radical adaptations [25]. For example, the Turkish agglutination "*evlerden*" can be seen as the composition of a stem and two word elements, "ev-ler-den", meaning "from the houses", composed by a concatenation of morphemes translating literally to "*house*-(*plural modifier*)-*from*". Clearly, simple white-space tokenization will not suffice in the recognition of these three morphemes.

Modern sub-word tokenization strategies, as discussed, put the linguistic significance of tokens aside. Tokens in the vocabulary are instead selected using model-based approaches that require an appropriate amount of training data but do not rely on explicit language-specific knowledge. In other words, these tokens are not seeking to have a one-to-one correspondence to morphemes, and may also span through different words, depending on the co-occurrence of character sequences in the training corpus.

*Sub-word segmentation*. As mentioned, traditional tokenization procedures often approximate linguistic units as an acceptable compromise. Modern segmentation procedures, on the other hand, often do not have explicit linguistic motivations or explanations and are instead based on automatic learning processes, trained for efficient tokenization on large unlabeled corpora. Unsupervised word segmentation with neural models has seen particular interest in languages that are notoriously difficult to segment because of their lack of white-space delimiters (Chinese, Japanese) or because of their highly productive morphologies tokens (Arabic, Hebrew) [26]. Reducing the number of OOV terms is particularly important for the latter case, as downstream tasks such as classification would incur too high a loss of information if they were just removed. However, it has been argued that languages such as these, as well as agglutinative languages, may be better served with character-level models or small sub-word inventories [27, 28], even though sub-word segmentation has reasonable motivation [8]. Cases like these reinforce the notion that there is no single best solution for language segmentation.

*Maximal decomposition*. As previously mentioned, some recent proposals have proposed maximal decomposition of text based on its underlying bytes rather than typographical tokens (e.g., words, sub-words, characters). An example widely used in recent models is that of byte-level BPE, which applies the BPE compression algorithm on bytes rather than characters [15]. This is not only a compact representation (up to 256 possible values for a vocabulary), but crucially agnostic to languages, and has seen success in languages particularly difficult to segment. Encoding byte-level representations is not however as simple as it may seem, as byte sequence representations are often much longer than character sequences. Moreover, as Mielke et al. [8] point out, byte-level modeling is not necessarily unbiased; while characters are intrinsically tied to language representation, different character encodings are unrelated to linguistics. For example, Unicode-based representations were not created with linguistic motivations, and different languages may have different representations (for example, might require multiple bytes per character). Another approach being explored is that of "visual" modeling, utilizing the pixels that compose the graphical representation of text, which may be promising for languages with rich visual features (e.g., Chinese, Korean) [29].

*Shared vocabularies*. Many NLP applications must be able to handle text in different languages simultaneously. It is possible to utilize a number of language-specific tokenizers, but shared vocabularies have also been proposed for multilingual systems. These systems work with a vocabulary composed of a variable number of word segments derived from different languages. Thus, there is no language-specific set of recognized tokens, but only an expanded multilingual vocabulary. As can be expected, a same token might be shared across different languages: in this case, its vectorial representations will have to encompass its meaning in multiple languages. While the sharing of learned representations is enticing, inconclusive results have been found in this regard [30]. Moreover, recent language representation approaches based on shared-vocabulary tokenization tend to be biased towards high resource languages such as English (even when oversampling low-resource languages), propagating this bias to downstream tasks such as TC [31].

A related work by Rust et al. [7] evaluated the performance of several monolingual tokenizers pre-trained on monolingual corpora and reports results in terms of two custom-tailored metrics. In their research, which concerns the effect of tokenization strategies on downstream tasks when paired with recent approaches, they explore the difference in performance between monolingual and multilingual tokenizers. The former are based on prior research on monolingual models and are mostly based on the WordPiece algorithm. However, many of these tokenization strategies rely on additional language-specific preprocessing. Examples include Japanese utilizing a pre-built morphological parser whose tokens are then split into characters, Arabic testing pre-segmentation techniques before applying the WordPiece algorithm, and Korean introducing bi-directional conditioning in the WordPiece algorithm. The authors found that the multilingual tokenizer performed inconsistently across languages, producing a lower number of tokens in morphologically poor languages and over-segmenting the richer ones. The latter are more challenging because root morphemes are frequently enriched with affixes to match the context of the sentence, including grammatical gender, case, number, or person. This translates to a higher number of possible combinations of words that require either more data or language-specific tokenizers. The issues and challenges faced by multilingual tokenizers and the shared vocabularies they produce are an active area of research [8].

*Summary*. Text segmentation is a fundamental part of any NLP task, with high linguistic relevance and important ramifications throughout the pipeline of a classifier due to its intrinsic ties to the embedding creation process. The previously mentioned work by Rust et al. [7] reports that the tokenization strategy (and, relatedly, the size of training data) are among the greatest driving forces for downstream task performance. They also found that utilizing monolingual tokenizers in multilingual models can lead to improved performance in most tasks and languages.

There is much more to be said about this topic, and we point interested readers to the work by Mielke et al. [8] which provides a comprehensive dissertation on the issues of tokenization strategies and emphasizes the limits of fixed vocabulary data-driven tokenizers, including the ones related to bias in data and language fairness for multilingual models.

## Other preprocessing operations

In this section, we briefly outline other common preprocessing operations applied to already tokenized text. Notably, most modern tokenizers already apply a number of the noise removal and "soft" normalization processes we will describe (e.g., lowercasing). Other more "destructive" operations, which remove or alter words altogether, should instead be considered carefully and on a case-to-case basis, as modern NLP models are typically trained to extract context from grammatically and morphologically sound sentences and performance will likely

suffer if they are applied to heavily preprocessed corpora with a very different distribution of words.

**Noise removal.**   The set of tokens produced by tokenization might contain unnecessary or misleading elements, such as superfluous symbols or characters. *Noise removal* refers to the set of operations used to remove those tokens and words that are deemed unnecessary or harmful to solve a specific task. Such procedures may also include lowercasing, misspelling correction, and standardization of slang words and abbreviations, which are all intended to reduce the number of different elements to be projected in the feature space. Earlier approaches commonly resorted to the removal of stopwords, non-informative words with no discriminative importance for classification and that are common in languages (e.g., articles, pronouns, etc.) [32, 33].

**Stemming and lemmatization.**   As traditional text interpretation approaches are unable to capture significant semantic information about words, a further simplification of the feature space can (and has been shown to) be beneficial [34, 35]. Therefore, simplifying words by reducing inflections to a common form can be helpful in relating words that earlier methods would otherwise be unable to tie together (e.g., "child" vs "children"). This is most commonly achieved through either *stemming* or *lemmatization*, which derive the stem or lemma (canonical form) of a word, respectively.

**Linguistic considerations.**   Many relevant linguistic aspects were already covered when discussing tokenizers and language segmentation in general, which is easily the most influential preprocessing step. Other operations, such as stemming and lemmatization, also have a similar linguistic connotation, but in a more traditional sense; indeed, they are generally rule-based or vocabulary-based, meaning that they are specifically created to process text in a language and depend on a manually defined set of rules and common affixes, stopwords and base lemmas. For example, the SpaCy rule-based lemmatizer uses a set of cascading rules that reduce tokens to a base form, according to possible word-forms that are applicable to the recognized PoS. The language-specific vocabulary is then used to determine if the lemmatized word actually exists.

Each language requires specific adaptations of rules and vocabularies to perform noise removal operations, which can notably have varying success in different languages. The reasons behind these differences in performance between languages are likely to be attributable to differences in the complexity of morphology that are more difficult to model through a set of rules. Additionally, vocabularies for rule-based procedures might potentially be incomplete because of the high variance in the number of lemmas in different languages.

For instance, lemmatizing a document in Greek is likely to be much harder than lemmatizing a document in Italian; as an empirical example, the SpaCy documentation reports a much lower score for the former's lemmatization accuracy [36, 37].

## Text representation

Following a preprocessing procedure, a body of text will be transformed into a list of separated, standardized tokens that might have been through multiple filters. Before it can be understood by a computer, however, it must be expressed in numeric form. Feature representation techniques are a fundamental part of any NLP application, many times trumping in importance the actual classification step of the overall pipeline. In this section, we give a brief overview of the most frequently utilized traditionally, and segue to a discussion of recent approaches.

**Language modeling.**   An important concept in text representation is that of *language models* (LMs). These are a statistical representation of text which has been studied for decades, though it has seen a new rise in popularity due to the application of deep neural models.

Intuitively, language models aim to predict the likelihood of a string given a preceding or surrounding context (usually a sequence of words, or, more in general, tokens). The related task is referred to as *language modeling*.

Formally, a statistical language model can be described as a probability distribution over sequences of words. Given a sequence of words $s = w_1, w_2, \ldots, w_m$, the model assigns a probability $P(w_1 w_2 \ldots w_m)$ to the whole sequence. While the goal is to assign probabilities to whole sequences of words, the task is related to that of computing the probability of an upcoming word and is framed as such. *N*-gram models are a simple example, making use of the Markov assumption, by which the prediction probability is based only on the last *n* words before it— i.e., $P(w_1 w_2 \ldots w_n) = P(w_i | w_{i-n} \ldots w_{i-1})$. Traditional algorithms such as Maximum Likelihood Estimation (MLE) [38, 39] can be used to solve the probability prediction task. The probability scores given are context-specific (in general, they relate to a better-structured sentence, such as a good translation).

## From text to vectors

We introduce in this section the most influential strategies for the representation of text sequences. We start by outlining the more traditional approaches for the conversion of text to numerical form, which are based on word occurrence frequency, and move on to the more advanced methods which utilize the underlying idea of language modeling.

**Bag-of-Words.** Traditionally, the most basic representation of text has been that of *Bag-of-Words* (BoW) [40–42]. As the name suggests, this model reduces bodies of text to unordered collections of words in which sentence structures and semantic relationships between its elements are ignored (hence, the intuitive visualization as a "bag of words"). Though simple, this approach has been widely used throughout ML applications (even outside of NLP, where it is commonly referred to as "Bag-of-Features") [41, 43–45]. Furthermore, it is common to utilize a feature extraction technique such as Term Frequency (TF), which maintains the relative frequency of words in a single vector for the entire text rather than a one-hot encoding of each word. This is usually paired with an Inverse Document Frequency (IDF) [46] factor, which penalizes common words within the entire corpus of texts (since they do not help discriminate between them). Vocabularies generated by TF-IDF representations may encounter time and memory complexity issues. One possible solution is to limit the maximum number of features represented (in practice, pruning low-scored words) or, alternatively, a dimensionality reduction algorithm can be applied. Popular approaches which have seen success include Principal Components Analysis (PCA) [47], Linear Discriminant Analysis (LDA) [48] and Non-Negative Matrix Factorization (NMF) [49].

**Word embeddings.** Earlier methods focused on capturing the syntactic representations of words but lacked the capability of encapsulating semantic meaning inferred from context. For example, they possessed no way to assimilate word synonyms. In the last decade, researchers have proposed to leverage language modeling to produce *word embeddings* as a solution to this problem. Intuitively, this self-supervised feature learning technique is aimed at learning a mapping between each piece of text (most commonly words, hence the name) to a *n*-dimensional vector of real numbers. These approaches are based on shallow neural networks, which learn these mappings through different learning procedures; in general, they are based on the assumption that the meaning of a word can be extracted from its surrounding words in a sentence. Some of the most popular and effective word embedding techniques based on these principles are Word2Vec [50], GloVe [51] and FastText [52].

Differently from BoW representations, which are only concerned with word occurrence statistics, this latter technique can embed much more information in the learned representation,

depending on the objective of the training procedure used to generate it. In the simplest case, word embedding techniques produce representations based on the surrounding tokens: similar contexts produce similar embeddings. This is generally the idea behind pre-trained embeddings, that are released for general usage and are not designed for specific tasks.

However, it should be noted that many strategies can be devised to enrich learned embeddings with more discriminative features, sometimes by fine-tuning pre-trained ones on other tasks. For instance, Qin et al. [53] propose a different approach, which uses two neural networks to learn features from randomly initialized embeddings. Features extracted by the first module are projected to the orthogonal direction of their counterpart, in order to learn more relevant, uncommon features. This can be seen as similar to the idea used to generate contextualized embeddings, which we describe in the next section.

**Deep language models.** Word embeddings represent an important milestone toward the creation of neural language models. As said, shallow learning-based architectures such as Word2Vec focused on the embeddings rather than the model itself, creating largely "static" (i.e., context-independent) vectorial representations. The denomination of embeddings as static can be attributed to the fact that polysemous words (words with more than one meaning) map to a same embedding, which can therefore be understood as a combination of the multiple senses of such word the model has encountered during the training process. From a practical point of view, these embeddings work like lookup tables, where every recognized word is mapped to a single fixed-size vector that sums up all of the contexts that a particular word has appeared in during training. Hence, for a given word to be encoded, the output vector is the same, no matter its context in the sentence.

A variety of deeper architectures have been applied to TC using pre-trained word embeddings, in an effort to improve the models' capacity and to create more meaningful semantic representation. Among other enhancements, Autoencoder frameworks and Recurrent Neural Networks have been particularly influential. The latter are prime candidates in the modeling of sequential data, and they have been applied with success to word embedding techniques [54, 55].

*Contextualized embeddings.* As more complex and deep architectures were applied to improve the learning process of text representation, some researchers proposed to use deep NN-based LMs to add context to word embeddings. This contextualization process is an augmentation of the previously described "static" embeddings. To obtain a *contextualized embedding* for a word, the static one is passed through a model that transforms variable-length sequences of left- and/or right-context words into a single fixed-length vector. Hence, unlike previous static word vectors, these embeddings are generated from both the static ones and the parameters of a contextualizer LM, producing distinct embeddings even for the same words used in different contexts [56]. The relation between contextualized embeddings and static embeddings is shown in Fig 1. In particular, note how the embedded sentence enters the model in its entirety, allowing the model to contextualize individual representations based on the surrounding tokens. Studies have shown that layers in deep language models are specialized to capture different linguistic information [57].

While this approach was first tried with recurrent-based LMs, notably ELMo and ULMFiT [58, 59], it has been rendered ubiquitous by the introduction of purely attention-based models, made popular by the seminal Transformer architecture [60] and the subsequent development of the Bidirectional Encoder Representations from Transformers (BERT) [61] and the Generative Pre-trained Transformer (GPT) [62]. Among other advantages, such deep models are able to benefit from increasingly large numbers of parameters, usually achieved by multiplying the number of layers in their architecture, something which was crucially not the case for recurrence-based models [63]. These LMs, which we informally term *contextualized LMs*, are hence

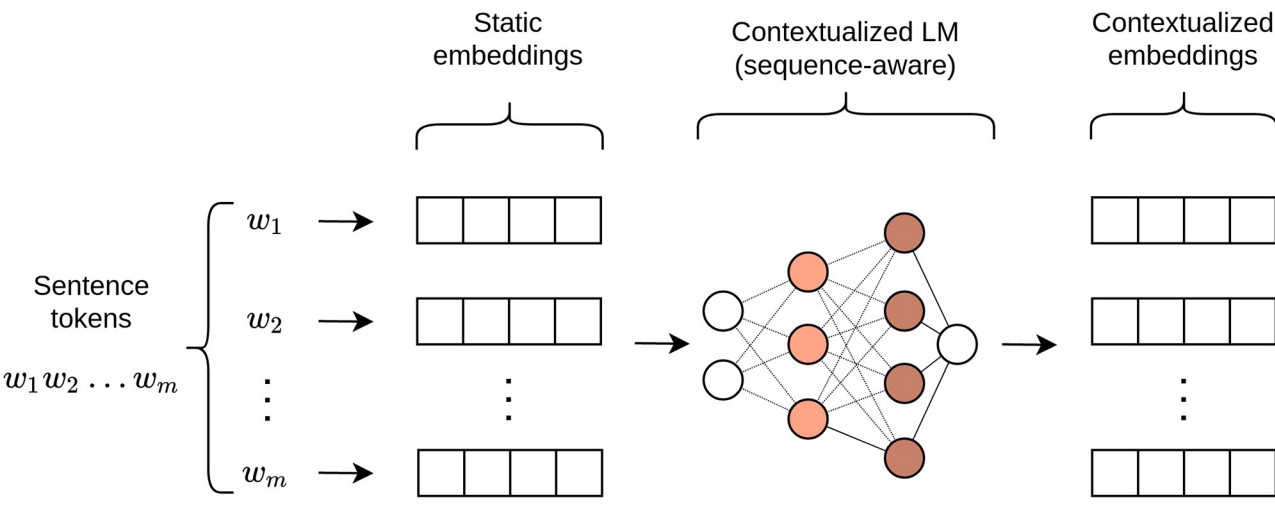

**Fig 1. Sample generation process for contextualized embeddings.**

able to disambiguate polysemous words by looking at the surrounding tokens in the sentence. Conversely, one should note that contextualized embeddings are not meant to be extracted "statically", i.e., as with a one-to-one mapping from word to vector. Instead, the language model should always be provided the surrounding context of a word in order to produce a meaningful word vector.

Contextualized LMs are usually pre-trained on a language modeling task (e.g., next word prediction) and are used as transfer-learning methods in other NLP tasks [64]. Adaptation to tasks is typically carried out through fine-tuning of the model, or part of it, on domain-specific data. Various strategies have been proposed, depending on the base model, one of them being training the backbone model with a task-specific head on top of it, possibly even freezing the backbone parameters.

## Feature extraction in other languages

In this section, we contextualize text representation techniques to their utilization in languages other than English, highlighting the challenges of training LMs and closing the section with an overview of models trained on Italian corpora.

**Traditional approaches in other languages.** As earlier approaches were not capable of expressing the semantic meaning of words, they can be largely seen as "detached" from language (with most linguistic aspects falling on tokenization approaches, as discussed). Therefore, the performance of methods such as BoW or TF-IDF relies largely on preprocessing and principled usage of statistical methods. Since not meant to really understand languages, their utilization does not see much or any difference when used on different ones (though their performance might vary depending on their specifics). In contrast, word embeddings are pretrained on large, usually monolingual corpora, and are thus specific to the language represented within the data.

Popular examples of such corpora are Wikipedia dumps [65] and the Common Crawl archive [66], whose size allows for more robust and generic representations. It is possible to manually train these embeddings from scratch (provided that the dataset is of sufficient size) or fine-tune a set of pre-trained ones; both approaches aim to enrich vectors with dataset-specific knowledge. Embeddings specialized on the domain data could (and usually do) result in better performances in downstream tasks such as TC.

It is hard, however, to determine how much data is required to meet the "sufficient size" criteria, since it specifically depends on the task at hand and the quality of the data. This adds to the fact that learning word embeddings is a long and computationally expensive procedure. Because of this, it is common to utilize pre-trained, open-sourced embeddings as a starting point. In the context of mid-resourced languages such as Italian, pre-trained word embeddings can usually be obtained reliably. A lower-resourced language might have to resort to manual training of these embeddings, which may require non-trivial computational resources—a topic we will address in more detail in the following sections.

**Contextualized language models in other languages.** Transformer-based language models have revolutionized how NLP solutions are sought. Unlike its predecessors, this generic methodological approach is applicable to a wide variety of tasks, often needing very little work to specialize it towards the specific downstream problem. As mentioned, however, the majority of research is done in the English language, which is taken as a "good representative" for the applicability of its results. Adaptations to other languages are created at different speeds and degrees, as the development of contextualized language models is made difficult by their high computational complexity and necessity for large amounts of data.

In the following, we will illustrate how impactful these requirements can be for practical development, analyzing the challenges and possible solutions being developed, as well as going into detail about the resource landscape for the Italian language.

*Pre-training of contextualized language models*. Pre-trained language models based on deep learning need to be trained on large corpora of text in order to achieve a good generalization capability. For instance, the original implementation of BERT was trained on BooksCorpus (800M words) [67] and English Wikipedia (2,500M words). The authors emphasize the necessity of utilizing document-level corpora, such as to extract long contiguous sequences which lead to better generalization. Finding these types of resources is much easier in the case of English, but mid-resourced languages such as Italian usually have access to sufficient resources for pre-training on self-supervised tasks. Indeed, this particular challenge will affect more severely low-resourced languages, rather than mid-resourced ones.

Recent research has begun to take different directions when it comes to pre-training approaches, attempting to either specialize or generalize pre-training data. While not in the context of classification but rather that of summarization, Zhang et al. [68] showcase a higher performance when data and learning objectives utilized in pre-training more closely mirror the final task of the overall system. This is in contrast to the generic approach of other language models, which are in many ways agnostic to downstream applications in favor of generality. Conversely, the recent GPT-3 [69] model tries to leverage massive datasets and processing power to create a model generic enough to overcome the need for specialized approaches. In particular, the GPT-3 is meant to address the issue of small downstream datasets, showing promising results for approaches with low label rates.

Both of these developments reveal insights into what we can expect future challenges to be. In the first case, researchers have empirically shown better results with specialized data; as we will showcase in the section discussing our findings on task-specific datasets, that kind of data is not as easy to come by as general-purpose, task-agnostic text information. In the second case, the difficulties are tied to the vast computational expenses, which we will discuss in the next section.

Notably, research has also pushed towards methods that are able to generalize well between languages. XLM-R [70] is a RoBERTa-based model pre-trained on more than 2 terabytes of unlabeled corpora in more than 100 languages. The result is a multilingual pre-trained model suitable for fine-tuning on a variety of multilingual and monolingual tasks. The authors reported competitive performance with respect to monolingual models. The development of

multilingual and multipurpose language models suggests the possibility of future research possibly converging towards fewer, more inclusive contextualized language models. Nevertheless, all language models gain much from the massive size of the datasets they are trained on, and the availability of such corpora is still problematic for under-represented languages. Moreover, we have highlighted how language-specific tools such as monolingual tokenizers may still be beneficial to downstream task performance, questioning their one-to-one replacement with purely multilingual approaches.

*Computational resources*. The computational resources required to develop contextualized language models of the BERT and GPT families are, without a doubt, incredibly high. Many recent evolutions of these models that have been proposed have tens of times the number of parameters of the original ones. While not a linguistic challenge per-se, it is evident that the conspicuous computational requirements will also act as entry barriers, preventing widespread research in this area.

While many authors do not disclose the actual training times and hardware infrastructure utilized by these models, it is safe to estimate that the upward trend in processing power required will continue [71]. The larger the models, the higher the number of trainable parameters for the network, translating into often prohibitive costs of development. Computational complexity is clearly challenging because of multiple aspects, spanning from environmental concerns (tied to the amount of power consumed to produce models that are substituted every year) to the fact that it is becoming more and more prohibitive to perform proper experimentation because of the cost of a single training procedure.

To put it into perspective, it is sufficient to consider the aforementioned GPT-3 model, whose largest iteration flaunts close to 175 billion parameters, which amounts to more than 1500 times the trainable parameters of BERT's base model. Researchers have estimated a positive correlation between number of parameters and model performance [72], theoretically justifying the push for larger and larger models. More recently, GPT-3 has been surpassed in size by even bigger models, such as Google's GShard [73] and Switch-C [74], which have 600 billion and 1.6 trillion parameters respectively. Fig 2 shows a visual representation of this trend in recent models.

While the examples provided are not representative of all Transformer-based language models, the lack of computational resources can be problematic even in more common scenarios. An example can be made of the fine-tuning procedure, which needs to be performed for any downstream task; however, working with such large models—even if pre-trained—might still be challenging just because of how expensive it is to load them into memory. Again, this severely limits the possibility for experimentation and evaluation in different languages.

*Reducing the cost of Transformer-based LMs*. In response to this issue, researchers have devised much more compact models which are still able to achieve similar results, while being considerably more applicable in practice. DistilBERT [75] leverages knowledge distillation to reduce the size of BERT models by up to 40% while retaining 97% of its effectiveness. TinyBERT [76] extends knowledge distillation to the task-specific learning stage. A different approach is proposed by models like ALBERT [77], which introduces parameter-reduction techniques to reduce the memory consumption and increase the training speed of BERT models. Similarly, ELECTRA [78] introduces a more sample-efficient pre-training task in place of masked language modeling (MLM), namely token detection.

In practical scenarios, whenever processing power is limited, utilizing downsized models such as these might be a solution. This is especially true for models such as ALBERT and ELECTRA, as they devise clever ways to improve the efficiency of the pre-training task, while DistilBERT and TinyBERT still require the original model as a "teacher" in the distillation process. It is also possible to perform a fine-tuning procedure without involving the language

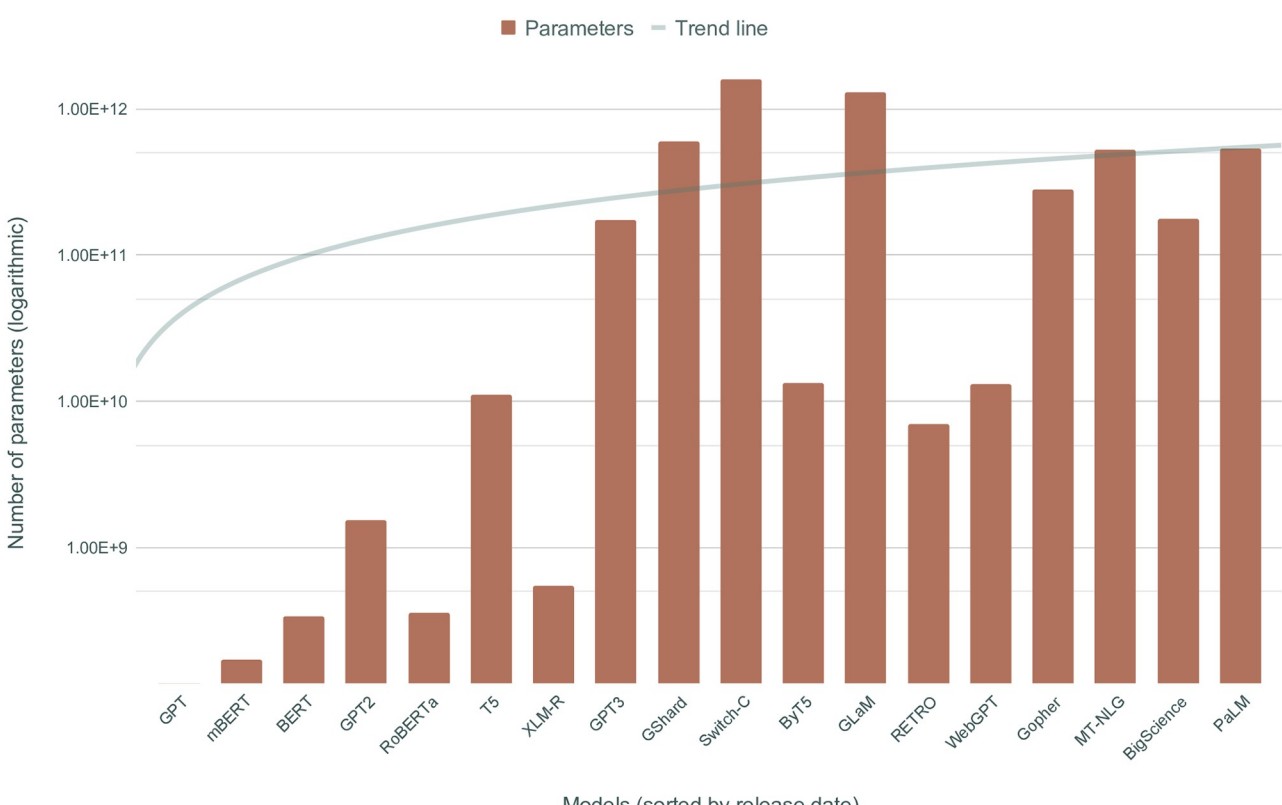

**Fig 2. N. of parameters (log scale) in the largest version of recent contextualized LMs, ordered by release date.**

model in the learning process (i.e., "freezing" the base model's weights). This is more akin to utilizing the underlying word embeddings in their agnostic state (but still contextualized). The computational resources necessary are therefore vastly reduced, though this severely shrinks the learning capacity of the overall system.

**Language models in Italian.** With the revolution brought about by BERT-derived models and their successors, researchers have quickly begun to study their applicability in specific tasks. In this section, we highlight a few of the studies made for Italian, as well as some multi-lingual approaches.

Tamburini et al. [79] studied the performance of BERT-like models in classic NLP tasks, such as PoS-tagging, NER, and universal dependency parsing, as well as some considerations on sentiment analysis. They highlight the most prominent pre-trained models available at the time of writing and find them to allow for a large increase in performance for almost all of them. An example of these works is ALBERTo [80], an Italian model based on a slightly modified BERT architecture and trained on tweets for sentiment analysis tasks. GPT models have also been adapted, with works such as GePpeTto [81], a GPT-2 based model for Italian—though it is evaluated on generative tasks rather than classification.

As was briefly mentioned, multilingual approaches have also been studied, mostly on Multi-lingual BERT [61], a LM trained on the concatenation of monolingual Wikipedia corpora from 104 languages. Pires et al. [82] devise a zero-shot cross-lingual model transfer, in which the model is fine-tuned for a downstream task in one language and tested for that same task in a different language. The results of their experiments demonstrate that the model is able to generalize to different languages (including Italian) quite well, though it performs best on

typologically similar languages. Nevertheless, the best performance is still achieved by fine-tuning on the target language, hence suggesting that it is preferable when possible. While multilingual adaptation is possible, authors argue that deeper fine-tuning is needed when compared to monolingual approaches, especially whenever the task is more related to semantics [82, 83].

*Existing models in Italian.* Through platforms like Hugging Face [84] and Tensorflow [85], pre-trained language models based on various architectures are made available for multiple languages. As it was for previous word embeddings, it has become common practice to open source such models because of how long and expensive their training procedure is. Whenever the computational resources are not available, it becomes necessary to rely on the contributions of others, which may not be as plentiful in all languages. Table 2 showcases some of the pre-trained models that are available for the Italian language at the time of writing. Minor changes like case sensitivity or vocabulary size differences are excluded, while the annotation "M" stands for "multilingual". Whenever the "# of parameters" column has multiple entries, it refers to the various model sizes available (usually, a smaller "base" model and a "large" one).

Theoretically, better results are to be expected if the domain of the downstream task (e.g., news articles) is contained in the pre-training dataset; however, this is an aspect that is becoming less and less important, as large amounts of data usually yield better results regardless. Pota et al. [83], who analyze the performance of various models on a Twitter sentiment analysis task, report that a generic BERT model pre-trained on large, general-purpose corpora of plain text can outperform a model pre-trained entirely on tweets like ALBERTo, even though the latter is trained on a corpus that is closer to the one used in the final task. The authors attribute this result to the size difference between pre-training datasets.

**Summary.** The state-of-the-art approach to the projection of text into a feature space involves the creation of contextualized language models, which must be utilized at inference time to extract context-specific embeddings on which to perform, for example, classification tasks. While not specific to classification, many studies have shown how this is the most important step of the pipeline, and therefore one that requires considerable attention. Recent trends have gone towards more and more costly models, which, in response, are largely open-sourced to allow practitioners with fewer resources to make use of these approaches. However, as shown, rigorous development and experimentation still require those resources in order to be performed, vastly limiting the possibilities for those who do not have such computing capabilities.

**Table 2. Italian pre-trained transformer models.**

| Name | Paper | Source | Architecture | # of parameters |
|---|---|---|---|---|
| Italian BERT | - | [86] | BERT | 110M |
| AlBERTo | [80] | [87] | BERT | 110M |
| Italian ELECTRA | - | [86] | ELECTRA | 110M |
| UmBERTo | - | [88] | RoBERTa | 110M |
| GilBERTo | - | [89] | RoBERTa | 110M |
| GePpeTto | [81] | [90] | GPT-2 | 117M |
| Recycled GPT-2 | [91] | [92] | GPT-2 | 117M / 345M |
| Multilingual BERT (M) | [61] | [93] | BERT | 172M |
| XLM-Roberta (M) | [70] | [94] | RoBERTa | 270M / 550M |

(M) Multilingual model.

## Classification step

So far, we have described preprocessing and feature representation approaches, fundamental to Text Classification but at the same time shared with a wide variety of NLP tasks. The importance of appropriate text representation cannot be understated; in fact, as previously stated, recent approaches have shown outstanding results with very simple classifiers, further cementing the notion that effective projection of text into an appropriate feature space is essential.

In this comparatively shorter section, we highlight the changes in how classification is tackled in traditional and recent approaches. Unsurprisingly, many end-to-end classifiers, especially neural ones, are largely based on effective feature representation, further supporting the idea that semantic understanding of the language is at the base of any NLP task. An informal —yet intuitive—explanation of this result is that understanding the content of a body of text is the most important step in the classification pipeline, much like a person would likely be able to label a piece of text if it understood what it meant.

### Traditional classification methods

Traditional learning models put a large focus on preliminary data preparation and feature engineering phases. While this is also true for modern models, earlier approaches required much more aggressive preprocessing, with a much higher dependency on the removal of noise and unimportant words that added no discriminative power to the pipeline. This can be challenging, as languages encompass a large and varied amount of rules of dependencies. Nonetheless, after a set of features has been extracted, it is possible to apply generic classification approaches. As they are generic, it is hard to attribute any real language-specific insight to them.

For the sake of completeness, we provide in Table 3 a high-level view of a number of traditional TC approaches. For a more in-depth description of these methods, we point to Kowsari et al.'s [5] survey. It is worth mentioning that these methods, still have a place in practical uses for TC—certainly in environments with small or very specific datasets, where injections of domain-specific knowledge in preprocessing steps and feature handcrafting may be relevant.

**Table 3. Traditional classification techniques.**

| Model | Advantages | Disadvantages |
|---|---|---|
| Rocchio Classifier [95] | Simple and computationally cheap | Lacks robustness, not well suited for multiclass classification or multimodal classes |
| Naïve Bayes [96] | Easy to implement and train, fast calculation process | Strong feature independence assumptions |
| Conditional Random Fields [97] | Flexible feature design, combining advantages of classification and graphical modeling | High computational complexity and issues with online learning |
| Hidden Markov Models [98] | Well-studied approach, suitable for sequentially ordered bodies of text | Strong assumptions typical of probabilistic methods |
| $k$-Nearest Neighbors [99] | Non-parametric, fast under the right conditions, easy adaptation to multiclass | Unfavorable scaling with high-dimensional spaces, choice of $k$ is arbitrary, a distance function between text bodies is hard to define |
| Support Vector Machines [100, 101] | Effective non-linear modeling even in high dimensional spaces, robust against overfitting | High memory complexity and requires a non-trivial decision of a kernel function, not transparent, does not produce probabilities directly |
| Decision Trees [102] | Naturally models categorical features, fast and interpretable | Very susceptible to noise and overfitting, weak against diagonal decision boundaries |
| Logistic Regression [103] | Easy to implement and train, does not necessitate re-scaling of features or fine-tuning | Strong independence assumption of data points, only suitable for linear problems |
| Random Forests [104] | Fast ensembling approach, reduces variance of single decision trees | Loss of interpretability and inference speed, still prone to overfitting |
| Ensembles [105, 106] | Collection of classifiers are more robust and accurate, less prone to overfitting | Expensive training, difficult interpretation and careful fine-tuning is required |

## Neural methods

The necessity for classical models to handcraft features has, over time, proven to be especially limiting. Due to the strong dependence between these features and the domain itself, good feature engineering often necessitates extensive domain knowledge. In turn, this makes approaches difficult to generalize to new tasks and languages.

The development of word embeddings therefore marks an important paradigm shift. Much of the work done by deep approaches is in fact towards automatic extraction of semantically meaningful representations from text. In this section, we provide an outline of how neural models, based largely on deep learning, have evolved in recent years, highlighting a trend where much of the focus is on text representation. This section does not aim at giving a comprehensive overview of the discussed neural architectures, as they are not the focus of this work. We refer to the surveys by Li et al. [4] and Minaee et al. [6] for a more comprehensive coverage.

**Multilayer Perceptrons.** In the earlier years of adoption of deep learning models, researchers developed deep neural networks based on simpler architectures, such as Multilayer Perceptrons (MLP), which displayed good results thanks to their ability to capture latent features automatically [107, 108]. Such models usually treat input text as an unordered Bag-of-Words, where input words are represented through some feature extraction technique (like TF-IDF or word embeddings). However, some of these approaches attempt to integrate further information about the syntactic structure of text, with examples such as Paragraph-Vec [108], which incorporates the syntactic ordering of words as well as the contextual information of paragraphs.

**Recurrent Neural Networks.** More influential, however, were architectures based on Recurrent Neural Networks (RNNs), as the ability to interpret text as sequences of tokens allows them to capture latent relationships between contextual words [109, 110].

In general, a simple RNN for text processing is fed a sequence of word embeddings, that are processed one at a time. At each time step, the model receives the next word vector and the hidden state of the previous time step. Standard RNN architectures are most frequently enhanced with more advanced gating mechanisms, the most popular being the Long Short-Term Memory (LSTM) [111] and Gated Recurrent Units (GRU) [112]. These enhancements address many of the gradient-related issues faced by vanilla RNN frameworks. The introduction of bidirectionality in RNNs has also been proven beneficial [113] and has been applied to LSTMs, with notable results such as ELMo [58], a language modeling approach that relies on BiLSTMs and is one of the first milestones in the development of contextualized word embeddings.

Among the most utilized approaches, encoder-decoders based on recurrence [54] have been particularly influential. The hidden layers of these architectures implicitly learn a semantically and syntactically meaningful representation of text that can be used for classification. On the downside, recurrent models have inherent limitations due to their sequential nature, as sequentiality precludes parallelization. Longer sentences can also run into memory constraints and, more crucially, are seen as RNNs true bottleneck because of how the network tends to forget earlier parts of the sequence, making for an incomplete representation [114].

**Convolutional Neural Networks.** Convolutional Neural Networks (CNNs), though most commonly used in the field of computer vision, have also seen applications in the context of NLP and TC [115]. The most straightforward application has convolutional filters applied over word embeddings, most commonly with size as wide as the embedding dimensionality, as to always consider the entire vector representation for each word. The main upside associated with CNNs is their speed and how efficient their latent representations are. Conversely, other

properties that could be exploited while working with images, such as location invariance and local compositionality [116, 117], make little sense when analyzing text. Many approaches have been proposed, one of the most popular being TextCNN [118], a comparatively simple CNN-based model with a one layer convolution structure that is placed on top of word embeddings.

**Graph Neural Networks.** In the last few years, graph representations have seen a resurgence in various fields of AI [119, 120]. In particular, Graph Neural Networks (GNNs) have received increasingly more attention [121], and this has also been the case for TC. GNNs utilize graph structures to capture dependencies and relations between their nodes.

Numerous well-established approaches to neural networks have been generalized to arbitrarily structured graphs. Among them, the convolution operation—usually applied to regular grid structures [122, 123]—is particularly popular because of its effectiveness and convenience [6]. Convolutions propagate information between nodes, and consecutive convolutions allow the network to spread the information further away, providing an effective way to model higher-order connectivity. Recently, successful approaches have been obtained on heterogeneous graphs in which nodes are both words and documents; TC is thus cast as a node classification task for document nodes.

The real strength of graph networks comes from their feature extraction capabilities, with examples such as TextGCN [124]. Both word and document embeddings are learned through convolutions. Researchers have also tried to train BERT and GCN models jointly, as in [125].

GNNs are among the few architectures able to compete with contextualized language models in downstream tasks and can perform quite well in low label rate datasets [124]. Some of the major weaknesses of graph-based approaches reside in model complexity, which becomes an issue with large-scale text corpora due to memory limitations. Simplified models such as SGC [126] help in this regard, while also mitigating one of the other major issues of GNNS, that of *oversmoothing*—where node representations converge to a same value and become indistinguishable [127].

**Transformer-based language models.** The already mentioned Transformer, proposed by Vaswani et al. [60], is considered the most recent major breakthrough in sequence processing methods and especially in NLP. The Transformer architecture is based on an encoder-decoder framework with multiple attentive blocks stacked together. Crucially, the main novelty resided in the removal of any recurrence-based layers for modeling sequentiality, instead relying on the *attention mechanism* alone. For further details, we point to surveys such as Gasparetto et al. [128] and Li et al. [4]. Transformer-based methods have built on the original architecture while maintaining the same basic principles, and have significantly boosted the performance achievable on various NLP tasks. During pre-training, these models are able to encode generic linguistic knowledge that can be transferred to any downstream task via a fine-tuning procedure on task-specific data.

Some of the most influential contextualized language models are based on research that suggests that limiting the architecture to either encoders or decoders may result in equivalent performance and lighter models [129]. According to this principle, the previously mentioned Generative Pre-trained Transformer (GPT) [62] utilizes an architecture based on stacking multiple transformer-decoder layers, resulting in an autoregressive model that is trained on a unidirectional next word prediction task. While particularly suitable for generative tasks, it has also been successfully adapted to TC. On the other hand, the seminal Bidirectional Encoder Representations from Transformers (BERT) [61] relies instead on a multi-layer bidirectional Transformer encoder architecture. This model employs specifically tailored learning tasks—in particular, masked language modeling (MLM) and next sentence prediction (NSP)—in order to incorporate bidirectional conditioning. The adaptation of BERT to

downstream tasks is very simple. In fact, outstanding results have been achieved in classification by simply fine-tuning a model that passes the representations obtained by the encoders through a single-layer, feed-forward neural network. It is common to allow this training procedure to also affect the representation learned by the pre-trained language models, such as to specialize the overall model on the domain of the task being faced. In practice, the changes to the language model parameters (i.e., everything that precedes the classification head) are minimal; this is desirable since if it were otherwise the language model would incur too great a loss of generality.

BERT and GPT laid the foundation for many variants and improvements to their original framework. Among them, we cite the Robustly optimized BERT approach (RoBERTa) [130], which explores the importance of hyperparameter choice and improves its learning procedure, and the GPT-2 [131], which instead improves mostly in terms of data utilized and scale of the models. More recent developments have also been proposed, both in terms of architecture and in scale (i.e., size of pre-training data and number of model parameters). See [128] for a more exhaustive coverage of the latest LMs. When discussing future research directions and challenges, we will highlight some of the most relevant to the topics of this survey.

## Summary of language factors in the TC pipeline

As mentioned at the start of this section, much of the focus of the classification pipeline has shifted towards effective text representation. Contextualized language models are able to perform exceedingly well across different tasks (including TC) with very simple classifiers—most frequently, a simple feed-forward layer. Crucially, researchers have reported that these results can be obtained even without large amounts of fine-tuning and parameter optimization, such as in the work by Tamburini [79], which studies these phenomena in the Italian language. We also found this to be true in our experiments with two Italian labeled corpora, as will be outlined in later sections.

We wrap up the overview of classification methods by drawing some conclusions on the overall classification pipeline, as viewed when considering different languages. We highlighted the importance of tokenizers; proper text segmentation is fundamental to the feature projection step and therefore has direct impact on the final downstream performance of tasks such as TC. Language-specific tokenization strategies have been shown to have advantages over generic, language-agnostic approaches. Still, many multilingual language models have relied on data-driven tokenizers, like BPE and WordPiece, achieving remarkably good results without being rooted in linguistic knowledge. An excellent example is that of ByT5, a recent multi-task transformer model which follows the byte segmentation approach previously mentioned and obtains state-of-the-art results [132]. Nonetheless, despite a few efforts in this direction, the impact of language-specific preprocessing on large language models has not been thoroughly explored.

We have also showcased how language modeling with contextualized representation has obtained outstanding results, further cementing the idea that effective semantic representations of text are arguably the most important phase of any NLP task. However, we mentioned how computation complexity can be particularly daunting. Utilizing pre-trained language models is certainly advantageous and effectively democratizes their adaptation to many downstream tasks, but may entail a certain "rigidity" in their adoption. For instance, it should be reminded that models such as BERT or GPT are closed-vocabulary; replacing the tokenization strategy is not possible without performing again the entire pre-training procedure. Therefore, testing how the performance of a downstream task is affected by changes within the classification pipeline is likely to be an expensive process.

Finally, we have highlighted how some researchers are experimenting whether models pre-trained with task-specific objectives and data can outperform general-purpose models. As of now, results seem inconclusive, and there is no clear indication of whether this approach (which is clearly much more complex) will be preferred to the generic approach.

## Datasets

The availability of annotated corpora is essential for NLP research. While the development of deep language models mainly leverages self-supervised strategies, labeled datasets are required for supervised tasks like TC. In this section, we provide a comprehensive list of resources available in two European languages and compare them with the resources available in the English NLP research area. We decided to focus our search on annotated corpora in Italian, which we regard as a mid-resource language, and French, which we instead consider a high-resource language.

While a consistent number of written English annotated corpora is available and heavily referenced in the literature, we find that the quantity of easily accessible resources in the languages we considered is still lacking (especially in Italian), limiting research on this theme. To reiterate, this is a fundamentally different issue from the one posed by low-resource languages, usually characterized by insufficient unlabelled data to even be able to perform self-supervised procedures, but it is by no means a less important one.

### Overview

**Text classification tasks.** We conduct a scientific literature search of annotated textual corpora presented or employed in research contributions. Reflecting Li et Al. [4], we focus on the following common TC sub-domains:

- *Sentiment analysis (SA)*: the task of understanding the emotional content of a piece of text, usually mapping it to predefined categories representing specific emotions. We include in this category the tasks of stance and polarity detection, as well as the identification of rhetorical devices, like irony, or linguistic properties, like subjectivity;

- *Topic labeling (TL)*: the task of extracting the topic (or theme) of a document, for example an article. This task is often related to content recommendation, since it can be used to map textual contents to user interests;

- *News classification (NC)*: classification of news into specific categories, like user interests or topics;

- *Question answering (QA)*: extractive question answering can be framed as a classification problem where the model, given a list of candidate sentences extracted from text and a target question, must decide which sentence contains the answer;

- *Natural language inference (NLI)*: given a pair of statements, the task is to determine if one is entailed by the other;

- *Named entity recognition (NER)*: the task of locating and classifying named entities mentioned in unstructured text into predefined categories;

- *Syntactic parsing (SP)*: the task of predicting the morpho-syntactic properties of words in sentences, like part-of-speech (PoS) tagging, speech dependencies, and semantic role labeling.

These tasks can be adapted to different domains, and many sub-tasks with different formulations are possible. They are commonly used as benchmarks in NLP research, especially as

part of multitask natural language evaluation initiatives, like the General Language Understanding Evaluation (GLUE) benchmark [133].

**Search criteria.** In order to balance search time and effectiveness, our search strategy is composed of three steps:

1. Search for datasets on Google Dataset Search (https://datasetsearch.research.google.com);

2. Search for publications on Google Scholar using keywords along the lines of "Italian text corpus" and "Italian Text Classification".

   a. The first two pages of results sorted for pertinence are explored;

   b. The same is repeated by filtering results based on their publication date, by looking at contributions published from 2019 onwards;

3. Search on PapersWithCode (https://paperswithcode.com/datasets) for contributions using the same keywords.

   For every publication, we explore all referenced publicly accessible data sources.

## Datasets in other languages

In this section we present the results of our search, omitting corpora that are not public or that are unavailable at the time of our search. We further filter out datasets with less than a few thousand labeled samples, unless they are highly specialized datasets that we deem potentially valuable for ML applications.

**Italian and French datasets.** We list monolingual corpora for Italian and French in Tables 4 and 5. Table 6 describes multilingual classification corpora containing one or both of these languages, and possibly others. We mark with a single asterisk (*) datasets available through a request for access. Additionally, we mark with (**) datasets that are not distributed for free or require specific affiliation. When defining tasks, we use the abbreviations introduced in the previous section and otherwise use TC to indicate a generic "Text Classification" task that does not fit any of the defined categories. For the sake of comparison, we give an estimate of the dataset size, based on the published documentation. Size can be expressed as the number of labeled sentences ("S"), tokens or words ("T"), or documents ("D") available in the corpus, and is comprehensive of all training, test, and evaluation splits. For multilingual datasets in Table 6, it refers to the number of samples in Italian and French (or, when similarly sized, an average of the two).

**English datasets.** A comprehensive list of English TC datasets is provided by Li et Al. [4]. In order to make a comparison, we report in Table 7 some of the most popular English datasets, along with their size and related tasks. In this specific case, our search is limited to popular datasets used within PapersWithCode recent contributions.

## Findings

Many of the Italian datasets listed in Table 4 were created for the EVALITA initiative, a periodical campaign organized by AILC for the evaluation of NLP and speech tools for the Italian language. The most recurrent tasks proposed in this initiative fall into the sentiment analysis and syntactic parsing domains. While most datasets assembled by participants in this initiative are made openly available and provide great value to the Italian NLP research, they are often small in comparison to English datasets for the same task, and always fewer in number. SemEval is a similar international workshop for the evaluation of semantic analysis systems [240].

**Table 4. Italian datasets.**

| Name | Paper | Source | Task | Size | Unit |
|---|---|---|---|---|---|
| ABSITA | [134] | [135] | SA | 10,000 | S |
| SENTIPOLC | [136] | [137] | SA (irony, subjectivity) | 9,400 | D |
| ATE_ABSITA | [138] | [139] | SA, TL | 4,300 | D |
| AMI 2020 | - | [140] | SA (misogyny) | 9,900 | S |
| R-ITA | [141] | [142] | SA (stance) | 1,000 | D |
| ChroniclItaly | [143] | [144] | NER, SA | 8,600 | D |
| IHSC | [145] | [146] | SA, SP | 6,900 | D |
| HaSpeeDe | [147] | [148]* | SA, SP | 8,500/3,500 | D |
| SQuAD-it | [149] | [150] | QA | 61,000 | D |
| Fact-Ita Bank | [151] | [152]* | NER, SP, NC | 65,000 | T |
| FLaIT | [153] | [154]* | NER, SP | 1,500 | S |
| PAISÀ | [155] | [156] | SP | 250,000,000 | T |
| KIPoS | [157] | [158]* | SP | 200,000 | T |
| iLISTEN 2018 | [159] | [160] | SP | 22,000 | T |
| PoSTWITA | [136] | [161] | SP | 6,700 | D |
| TUT | [162] | [163] | SP | 3,500 | S |
| TE-EVALITA 2009 | [164] | [165] | NLI | 800 | D |
| GxG | [166] | [167] | TC (gender) | 11,000 | D |
| DaDoEval | [168] | [169] | TC (date) | 2,800 | D |
| AcCompl-It | [170] | [171]* | TC (acceptability, complexity) | 1,680/2,530 | S |
| ITAmoji | [172] | [173]* | TC (emoji prediction) | 275,000 | D |

* Available through a request for access.

**Table 5. French datasets.**

| Name | Paper | Source | Task | Size | Unit |
|---|---|---|---|---|---|
| French Twitter SA | - | [174] | SA | 1,500,000 | D |
| Alolciné | - | [175] | SA | 200,000 | D |
| French Sexism Detection | [176] | [177] | SA (sexism) | 11,800 | D |
| Event2018 | [178, 179] | [180]* | SA (stance), TC (event) | 15,000/137,000 | S |
| CAS | [181] | [182]* | SP, SA (uncertainty, negation) | 4,900 | D |
| E-FRA | [141] | [142] | SA (stance) | 2,000 | D |
| FQuAD | [183] | [184]* | QA | 26,000 | D |
| PIAF | [185] | [186] | QA | 3,800 | D |
| Quaero Broadcast News XT | - | [187]** | NER | 1,500,000 | T |
| Quaero Old Press XT | - | [188]** | NER | 1,300,000 | T |
| La Recherche | - | [189]** | SP | 447,000 | T |
| FTB | [190] | [191]* | SP | 644,000 | T |
| ParisParl | - | [192] | SP, TC (affiliation) | 203,000,000 | T |
| French Corpus MWE | [193] | [194] | SP | 166,000 | T |
| French FraCaS | [195] | [196] | NLI | 346 | D |

* Available through a request for access.
** Require payment or specific affiliation.

**Table 6. Multilingual datasets.**

| Name | Paper | Source | Languages | Task | Size | Unit |
|---|---|---|---|---|---|---|
| Webis-CLS-10 | [197] | [198] | Fr, En, +2 | SA | 69,000 | D |
| Amazon Reviews ML | [199] | [200] | Fr, En, +4 | SA | 210,000 | D |
| SemEval-2016 Task 5 | [201] | [202] | Fr, En, +6 | SA, TL (aspect) | 2,400 | S |
| Reuters Corpus Volume 2 (RCV2) | [203] | [204]* | It, Fr, +11 | NC | 28,406/85,393 | D |
| MLSUM | [205] | [206] | Fr +4 | NC | 425,000 | D |
| KB Europeana Newspapers NER | - | [207] | Fr +3 | NER | - | - |
| WikiAnn | [208] | [209] | It, Fr, +280 | NER | 7,5 mln | T |
| DAWT | [210] | [211] | It, Fr, En, +3 | NER, EDL | 1,5 mln | D |
| WikiNER | [212] | [213] | It, Fr, En, +6 | NER, SP | 260,000 | S |
| NewsReader MEANTIME | [214] | [215] | It, En, +2 | NER, SP, NC | 15,000 | T |
| Universal dependencies | [216] | [217] | It, Fr, En, +100 | SP | ~1 mln | T |
| XL-WiC | [218] | [219] | It, Fr, +1 | SP | 2,000/70,000 | S |
| Aranea | [220] | [221] | It, Fr, +20 | SP | 120 mln/1,2 bln | T |
| PANACEA | [222] | [223] | It, Fr, +2 | SP | - | - |
| MKQA | [224] | [225] | It, Fr, En, +23 | QA | 10,000 | D |
| CLEF QA Test Suites | - | [226]** | It, Fr, En, +7 | QA | 160,000 | D |
| XLNI | [227] | [228] | Fr, En, +13 | NLI | 7,500 | D |

* Available through a request for access.

** Require payment or specific affiliation.

Multilingual datasets provided for the proposed tasks tend to be small and, while access is provided, they are not easy to find and use outside the context of these initiatives.

We hereby discuss the availability of task-specific datasets as compared to analogous English counterparts (Table 7). One should note that some of the datasets presented, especially in French, are made available through the ELRA-ELDA catalog (available at http://www.elra.

**Table 7. English datasets.**

| Name | Languages | Task | Size | Unit | Reference |
|---|---|---|---|---|---|
| 20 Newsgroup | En | NC | 18,800 | D | [4, 229] |
| Reuters | En | NC | 10,700 | D | [4, 230] |
| R8 | En | NC | 7,600 | D | [4] |
| R52 | En | NC | 9,100 | D | [4] |
| RCV1 | En | NC | 804,000 | D | [203] |
| AG News | En | NC | 127,600 | D | [231] |
| TREC-6 | En | QA | 5,500 | D | [232] |
| SQuAD 2.0 | En | QA | 150,000 | D | [233] |
| Yahoo!Answers | En | QA | 1,460,000 | D | [234] |
| Yelp-2 / Yelp-5 | En | SA | 8,600,000 | D | [235] |
| Amazon-5 | En | SA | 3,650,000 | D | [4] |
| Amazon-2 | En | SA | 4,000,000 | D | [4] |
| IMDb | En | SA | 50,000 | D | [236] |
| DBpedia | En +124 | TL | 630,000 | D | [4, 237] |
| MultiNLI | En | NLI | 433,000 | D | [238] |
| CoNLL-2003 | En +1 | NER | 301,000 | T | [239] |

info/en). This resource requires an ELRA membership plan and/or the payment of a fee in order to be accessed.

**Syntactic parsing.**   Corpora for syntactic parsing (SP), like PoS-tagging and lemmatization, are well resourced in both languages reviewed in this paper and are featured in several multilingual treebanks (like Universal Dependencies [216] and Panacea [222]). Furthermore, the PAISÀ corpus stands out as a large monolingual dataset in Italian for these tasks.

**News classification.**   On the other hand, we noticed a lack of news classification (NC) datasets in Italian and French. The only notable exceptions are the MLSUM and RCV2 datasets. The MLSUM multilingual dataset contains news extracts labeled with their summaries and topic, and it is available in French but not Italian. On the other hand, the Reuters RCV2 dataset for multilingual news classification contains both Italian and French sub-corpora. This dataset can only be accessed by sending a request and signing an organizational agreement.

**Topic labeling.**   Likewise, topic labeling (TL) datasets are also scarce in Italian and French. Wikipedia dumps and DBpedia represent a remarkable source of crowdsourced semi-structured data that can be employed for topic labeling and TC in general and are available in hundreds of languages. However, labels must first be extracted and merged following some criteria which have not yet been standardized. Though these corpora have already seen use in the literature [241, 242], there is no consistent set of annotations to be used as reference. While categories assigned to Wikipedia pages by contributors are often used as predictive targets, these frequently contain spurious or improper information that can be treated in many different ways, or should arguably be removed entirely.

**Sentiment analysis.**   Multiple sentiment analysis (SA) datasets are available in Italian and French—often targeting user-generated content—for detection of polarity, political stance, or specific rhetorical devices (like irony). More than ⅓ of the Italian datasets are scraped from social or e-commerce platforms, especially Twitter.

**Question answering.**   There is at least one large question answering (QA) dataset for both Italian and French, as well as several multilingual ones which contain both languages. For this task, the size of these datasets is comparable to the main English QA datasets.

**Named entity recognition.**   Similarly, there are multiple large multilingual corpora for named entity recognition (NER) tasks, at least for the most classic formulation of this task aimed at recognizing "person", "location" and "organization" entities.

**Semantic entailment.**   We found two semantic entailment (NLI) datasets containing the Italian and French languages, the largest containing 1,000 and 7,500 samples respectively. By comparison, one of the most popular English NLI datasets (MultiNLI) is more than 50 times the size of the French one mentioned.

## Cross-lingual benchmarks

Multitask evaluation benchmarks like GLUE, SUPERGLUE (for English), and FLUE (for French) are increasingly popular tools to evaluate models across a wide variety of NLP tasks. This incentivizes models to share knowledge across different tasks and gain sufficient language understanding to generalize on a wide range of applications. Notably, the recent publication of the XTREME and XGLUE benchmarks introduced support to multilingual and cross-lingual cross-task evaluation. Still, for some tasks, not all languages are available. For example, among the classification tasks we are interested in, XGLUE supports only PoS-tagging (included in our SP category) and web page ranking on the Italian language. Additionally, they do not provide training data in every language, and, for some tasks, data is extracted from other multilingual datasets (namely XNLI, Universal Dependencies, and WikiAnn). Table 8 summarizes popular language-specific and cross-lingual benchmarks.

**Table 8. Linguistic benchmarks.**

| Benchmark | Paper | Languages | Tasks | Cross-Lingual |
|---|---|---|---|---|
| GLUE | [133] | En | QA, SA, TL, NC | |
| SUPERGLUE | [244] | En | TC, NLI, QA | |
| FLUE | [245] | Fr | TC, SP, NLI, PAR* | |
| XTREME | [246] | It, Fr, En, +37 | NER, SP, NC, QA | ✓ |
| XGLUE | [243] | It, Fr, En, +8 | NER, SP, NC, QA | ✓ |

* Paraphrasing.

The goal of these initiatives is not to provide more monolingual data for languages with fewer resources, but rather to encourage the evaluation of cross-lingual models capable of transferring knowledge across different languages, even those with little or no training data. Their contribution is important in that it provides a standardized evaluation environment for deep learning models that could alleviate the common low resource issue [243].

## Applicability evaluation

We have previously mentioned the rising computational costs of developing state-of-the-art NLP solutions. In this section, we simulate a practical case by synthesizing two custom multilabel classification datasets. In our research, we have found that multilabel TC in the Italian language (and, to some degree, in French) are understudied, in contrast to binary and multiclass TC. In a similar vein to Tamburini [79], our study is aimed at gauging how easily and how well these methods can apply to new tasks and datasets with a constrained amount of resources (i.e., only fine-tuning).

### Datasets utilized

We use one monolingual dataset per language for each studied task. We decided to tackle the multilabel classification task, as it was more interesting for our research work and, to some extent, is less documented in the literature. An empirical evaluation of the labels in our TL dataset finds that the categorizations utilized are overlapping and cannot be easily decomposed into binary sub-classification tasks (e.g., "sports" vs "winter sports") [247]. A similar consideration can be made of the NC dataset, which was multilabel by construction.

We chose to synthesize these datasets mainly because of the scarcity of other options in Italian. In the first case, we found that synthesizing a TL dataset from Wikipedia was the only way of obtaining a reasonably large, general-domain, annotated corpus in Italian. Similarly, the RCV2 dataset was chosen for the NC task because no other public collection of annotated news articles was available in Italian.

**Topic labeling.** We synthesized a dataset for the Topic Labeling task using Wikipedia dumps in all three languages. For each dump, articles and related topics are extracted using a modified version of the popular WikiExtractor tool (see https://github.com/attardi/wikiextractor). After an exploration of the data, we came to the conclusion that Wikipedia categories are ill-suited for a topic labeling task since they are often too specific and hardly provide a good topic indication. Therefore, we decided to use a different approach, and annotate extracted articles with the Wikipedia portals they are assigned to.

Currently, there are roughly 500 portals within the English version of Wikipedia, while there are more than 500,000 categories. Wikipedia itself states that portals serve as entry points for articles that belong to the same broad subject [248], thus making them better targets for

our task. Our final datasets contain only the 100 most populous portals, and article frequency has been limited to a maximum of 50,000 articles per label. This was done both to contain the dataset size and to reduce class imbalance.

**News classification.** For the News Classification (NC) task we utilized the Italian and French subset of articles in the Reuters multilingual collection (RCV2), as well as the English monolingual collection contained in RCV1. The English Reuters collection has been for a long time one of the staple TC corpora utilized for experimental purposes [247, 249], though its multilingual version has not seen as much attention. The articles are not "parallel", in the sense that they do not contain the same content in different languages, but are different articles altogether.

The RCV1/2 articles are labeled with a variety of tags that describe their content at varying degrees of specificity. The most consistent and interesting tags across languages were topic codes; within such codes, subjects are ordered in a hierarchical manner. However, articles are often tagged inconsistently—the depth of hierarchy within tags ranges between two and four levels, and documents are sometimes only partially tagged. We decided to retain topics at the second level of specificity; each article is tagged with all second-level topic codes it contains and stripped of any other. Only topics that had at least 500 representatives were included in the final datasets. Articles are deprived of all topics excluded this way, and the article is discarded if it contains no topics after this process.

**Analysis of datasets.** Final statistics of the described datasets are reported in Table 9. The main difference between the TL and NC datasets is the length of the articles; an average over the number of tokens per document reveals Reuters articles are comparatively much shorter than extracted Wikipedia articles. Indeed, Wikipedia articles are usually much more descriptive, while Reuters articles are presented in a very concise and to-the-point format. As the LMs we worked with allow for a maximum of 512 tokens as input, we can expect a truncation to be much more frequent in the case of Wikipedia articles. The information loss should however not be dramatic, as most discriminative information is usually found at the beginning, where the article is introduced.

Figs 3 and 4 depict the distribution of the number of topics per article for the ItWiki-100 and RCV2it datasets, respectively. The same statistics for the French and English counterparts are supplied in the S1 Datasets supplement. Unsurprisingly, most articles have few labels (between one and three), with a large amount having only one label. The larger Wiki datasets have a longer tail-shaped distribution, with a few outliers having a large number of labels, but that overall make up a small part of the datasets (for instance, articles with four or more labels make up less than 1.4% of the ItWiki-100 dataset).

We further report in the S1 Datasets supplement the distribution of topics, i.e., the number of articles labeled for each specific topic. All datasets follow a similar distribution, with a number of well-represented classes and a lower bracket of classes that are comparatively underrepresented. We point out that class imbalance was much more severe in the raw data we

**Table 9. Statistics on the examined datasets.**

| Name | Classes | Avg n. tokens | Samples | Task |
|---|---|---|---|---|
| ItWiki-100 | 100 | 354 | 892,573 | TL |
| FrWiki-100 | 100 | 362 | 1,494,761 | TL |
| EnWiki-100 | 100 | 741 | 329,626 | TL |
| RCV2it | 15 | 123 | 25,750 | NC |
| RCV2fr | 38 | 224 | 79,173 | NC |
| RCV1en | 57 | 216 | 758,149 | NC |

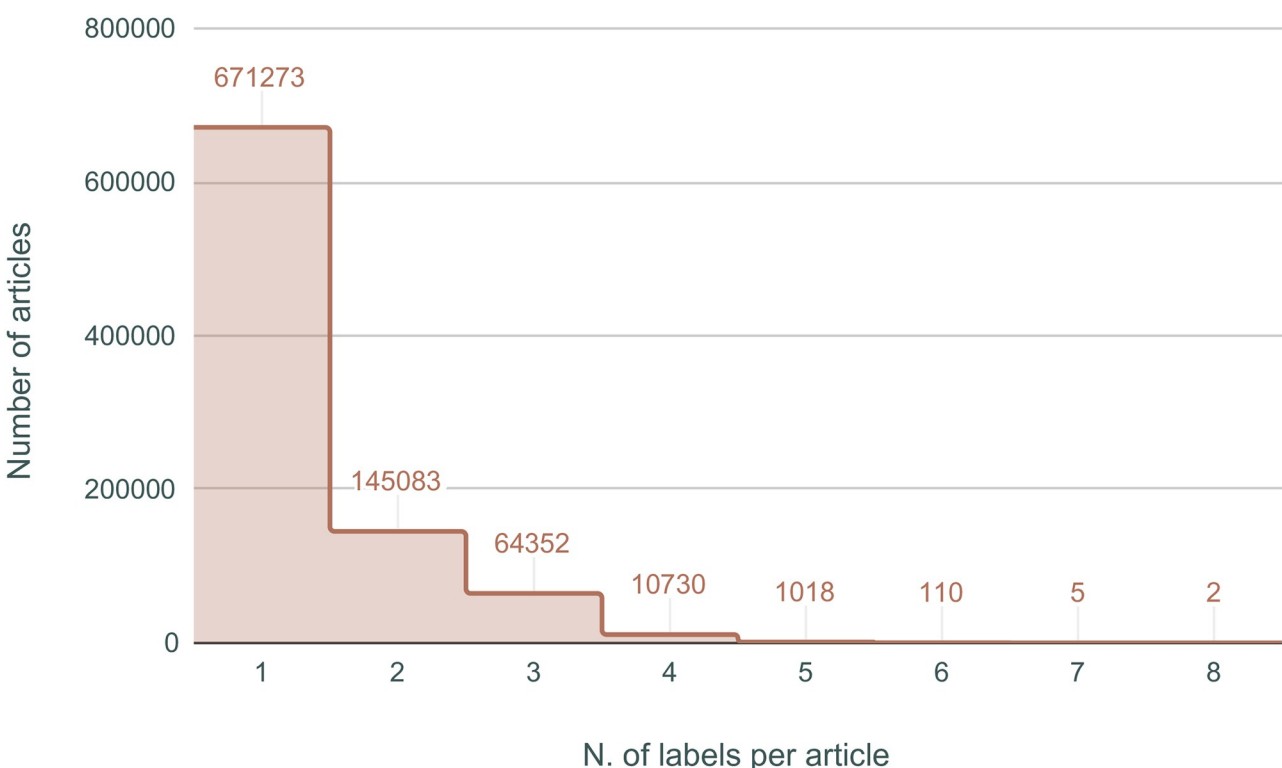

**Fig 3. Distribution of the number of labels in ItWiki-100.**

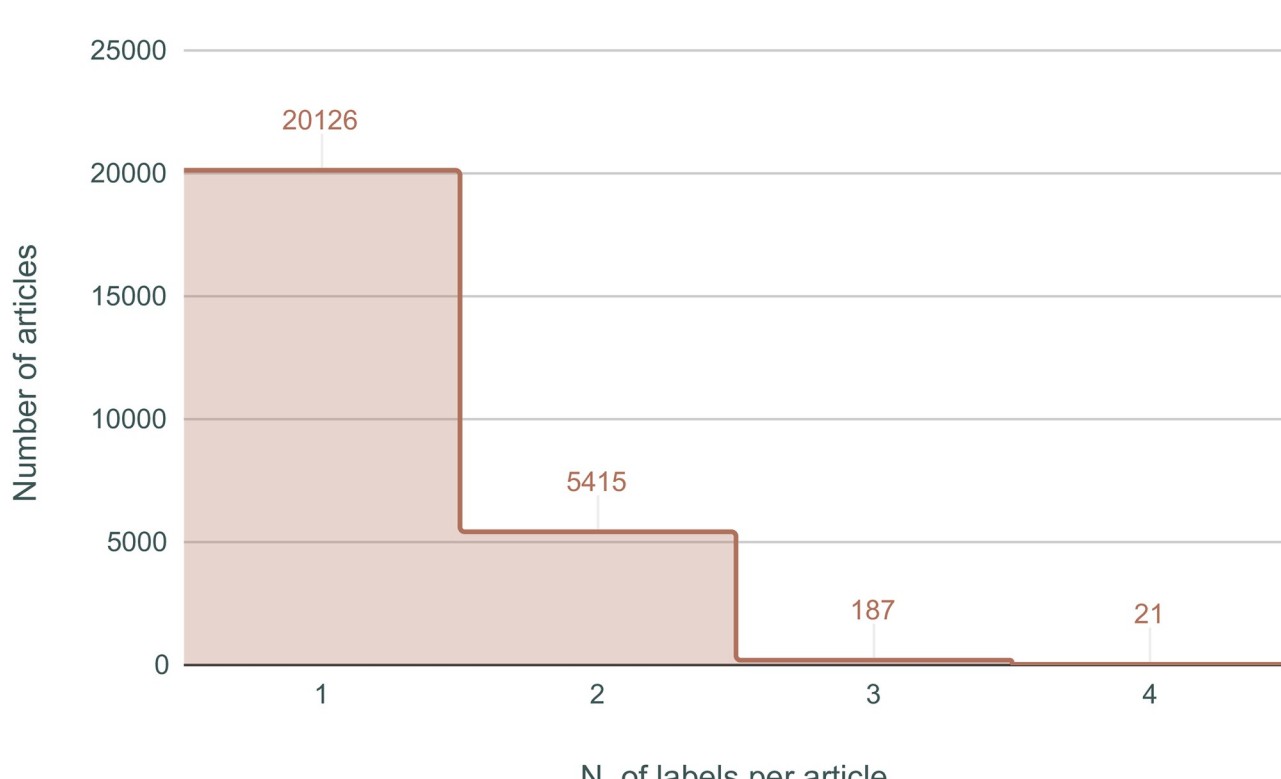

**Fig 4. Distribution of the number of labels in RCV2it.**

processed, with a much smaller number of dominant classes and a much larger number of topics with next to no representation. As it stands, the dataset still remains unbalanced, but in a way that, in our opinion, poses an interesting challenge.

## Experimental setup

For all methods, excluding FastText and classical approaches, the input documents are truncated, keeping only the first 512 tokens (or padded to that size). For training, every dataset is split into a training, validation, and test set: 40% of the data are reserved for testing, and 20% of the remaining samples are used for validation. Splits are produced in a way to preserve the distribution of labels, through a stratification strategy [250, 251]. Training was carried out on an NVIDIA GeForce RTX 2080 Ti GPU. More details on the training procedure are given in the S1 Appendix.

**Evaluation metrics.**   One of the most adopted evaluation metrics for multilabel classification tasks is that of F1-score, defined as the harmonic mean of precision and recall, as in the equation below.

$$F_1 = 2 \cdot \frac{\text{Precision} \cdot \text{Recall}}{\text{Precision} + \text{Recall}}$$

For multilabel and multiclass problems, it can be computed separately for each class and then averaged, obtaining the *macro* F1-score. In such a case, every class contributes equally to the final score, hence providing a more challenging metric for unbalanced datasets. On the other hand, a *micro* average reduction strategy is used when computing the metric globally with no weighting.

In our tests, we report the F1-score with macro-averaging across all categories along with the accuracy score, as the latter is an interpretable measure of the overall fraction of correct predictions. In its computation, the predicted set of labels must exactly match the ground truth for it to be considered a true positive (this score is sometimes referred to as "*subset accuracy*").

**Methods applied.**   We present quantitative results for an array of models that either are or have been state-of-the-art approaches to solving the task of TC. A more thorough analysis of the decision process behind these models is provided in the S1 Appendix. We start by providing a strong baseline with classical methods, of which we test Naïve Bayes and linear Support Vector classifiers as representatives. As examples of neural networks preceding the Transformer era, we showcase the results of FastText [252], XML-CNN [241] and a BiLSTM-based classifier. We then trained Transformer-based methods, using open-sourced models pretrained on language modeling tasks over large corpora. This last set of methods currently achieves the best results on the vast majority of downstream NLP tasks.

Every method is trained and tested 4 times per dataset, each time on a newly generated train and test split, and we list the final average metrics evaluated on the test set, along with the standard deviation over all runs, in Tables 10 and 11. As an exception, and because of technical constraints, we train XLM-R only once per dataset, as this model is much heavier than the already expensive BERT.

## Discussion on results

When it comes to classical methods, the results obtained are quite favorable. The one-vs-rest ensemble of linear SVCs proved to be the strongest baseline, while our application of multinomial Naïve Bayes lagged behind quite a bit (though it was considerably faster). Among

**Table 10. Test set macro F1 score for the tested TC methods.**

| | Italian | | French | | English | |
|---|---|---|---|---|---|---|
| Model | ItWiki | RCV2it | FrWiki | RCV2fr | EnWiki | RCV1en |
| Naïve Bayes (OVA) | 0.620 ± 0.000 | 0.765 ± 0.004 | 0.551 ± 0.001 | 0.661 ± 0.003 | 0.636 ± 0.001 | 0.563 ± 0.014 |
| Linear SVM (OVA) | 0.824 ± 0.000 | 0.796 ± 0.008 | 0.737 ± 0.000 | 0.724 ± 0.003 | 0.803 ± 0.001 | 0.717 ± 0.003 |
| FastText Classifier | 0.815 ± 0.001 | 0.767 ± 0.007 | 0.757 ± 0.001 | 0.641 ± 0.007 | 0.744 ± 0.054 | 0.696 ± 0.008 |
| BiLSTM (GloVe) | 0.836 ± 0.001 | 0.805 ± 0.002 | 0.769 ± 0.001 | 0.700 ± 0.014 | 0.812 ± 0.005 | 0.766 ± 0.007 |
| XML-CNN (FastText) | 0.827 ± 0.001 | 0.776 ± 0.009 | 0.789 ± 0.002 | 0.669 ± 0.011 | 0.782 ± 0.004 | 0.730 ± 0.007 |
| BERT (base) | **0.870** ± 0.001 | **0.840** ± 0.006 | **0.840** ± 0.001 | **0.768** ± 0.005 | **0.855** ± 0.002 | **0.781** ± 0.004 |
| XLM-R (base) | 0.868 | 0.836 | 0.832 | 0.739 | 0.846 | 0.772 |

Standard deviation over runs is reported ($\pm \sigma$). Best results are presented in bold.

**Table 11. Test set subset accuracy score for the tested TC methods.**

| | Italian | | French | | English | |
|---|---|---|---|---|---|---|
| Model | ItWiki | RCV2it | FrWiki | RCV2fr | EnWiki | RCV1en |
| Naïve Bayes (OVA) | 0.432 ± 0.001 | 0.629 ± 0.002 | 0.287 ± 0.001 | 0.475 ± 0.018 | 0.392 ± 0.007 | 0.473 ± 0.019 |
| Linear SVM (OVA) | 0.744 ± 0.001 | 0.717 ± 0.005 | 0.587 ± 0.001 | 0.656 ± 0.003 | 0.669 ± 0.001 | 0.677 ± 0.002 |
| FastText Classifier | 0.741 ± 0.002 | 0.678 ± 0.005 | 0.603 ± 0.001 | 0.611 ± 0.006 | 0.682 ± 0.053 | 0.670 ± 0.004 |
| BiLSTM (GloVe) | 0.763 ± 0.002 | 0.727 ± 0.007 | 0.637 ± 0.001 | 0.657 ± 0.008 | 0.680 ± 0.009 | 0.725 ± 0.005 |
| XML-CNN (FastText) | 0.764 ± 0.002 | 0.712 ± 0.005 | 0.661 ± 0.003 | 0.644 ± 0.008 | 0.666 ± 0.005 | 0.710 ± 0.001 |
| BERT (base) | **0.808** ± 0.002 | **0.773** ± 0.006 | **0.724** ± 0.002 | **0.696** ± 0.007 | **0.753** ± 0.003 | 0.735 ± 0.002 |
| XLM-R (base) | **0.808** | **0.773** | 0.716 | 0.688 | 0.743 | **0.740** |

Standard deviation over runs is reported ($\pm \sigma$). Best results are presented in bold.

preprocessing operations, we found that lemmatization and *n*-gram discovery did not produce significant differences in results, so we do not report their effect in the final tables.

Neural networks that predate the Transformer era showcased strong performances, usually surpassing traditional methods. In our experiments, exceptions were likely to be attributed to the size of training data. NNs had better results for larger datasets, giving instead ground to classical methods whenever training samples were more scarce. Moreover, on smaller datasets (like RCV2it), we observed a noticeable margin of variance between the results of different runs of said networks, which were instead very consistent on larger datasets (like our "Wiki-100" datasets).

For these models, we experimented with different pre-trained embeddings (Word2Vec, GloVe, FastText), and found that FastText embeddings gave the best result for XML-CNN. In the case of BiLSTMs, however, we found the best results were instead obtained with GloVe embeddings, despite their comparatively restricted vocabulary size. Furthermore, BiLSTMs benefited from the removal of stopwords in their input text, something that we did not find to be true in the case of XML-CNN. Nonetheless, the gap in the results between the two models was noticeable but not dramatic.

Unsurprisingly, the attention-based Transformer architectures outshined other methods on all our datasets. An important aspect of these models that warrants being reiterated is their ease of application to downstream tasks. In fact, only a few epochs of tuning were necessary to obtain these results. Even so, they were still the most computationally complex and required the longest time to fine-tune. While monolingual BERT models performed best, XLM-R

proved to have very strong performances, even though it is a multilingual model with a vocabulary diluted across many languages.

**Language-specific considerations.** On average, we observe that Italian models perform at the same level as (or slightly better than) English methods. French tasks, on the other hand, proved to be slightly harder on both datasets. In all cases, the trend of performance between methods is similar; as expected, contextualized language models perform the best across the line. We therefore focus this discussion on these models, are they are the most interesting to cover.

Many factors are likely to be influencing the differences in the reported results. First and foremost, the monolingual models were pre-trained on different corpora of different sizes. Diving into specifics, the Italian LM was pre-trained on 81GB and 13B tokens of data, taken from OPUS and OSCAR corpora [86]. The English BERT model is trained on the BookCorpus and Wikipedia dump (13GB), as outlined in the original paper [61, 245]. The French model is trained on a mixture of French documents, extracted from Project Gutenberg (a collection of e-books), OPUS, Wikimedia, and Common Crawl, amounting to 71 GB of data [245]. This latter model is also trained with a MLM objective only, while the others use both MLM and NSP, and it has more learnable parameters: 138 million instead of 110. Finally, XML-R was trained on 2.5TB of data in 100 languages, extracted from the Common Crawl corpus [70]. In it, the amount of data per language is variable: 300 GB for English, 30 GB for Italian, and 57 GB for French text. Another important factor to be considered is the difference in the size of our classification datasets. The RCV2 dataset contains a relatively small number of Italian articles when compared to both French and English. The number of target labels is also variable across languages, resulting in TC tasks that are likely to be on a slightly different level of difficulty.

As a consequence, given the conspicuous differences in pre-training, it is hard to make any consideration about the impact of the languages alone on the results. In this work, we aimed to give a generic overview from a linguistically inclusive perspective aimed at practical applications; indeed, we managed to obtain impressive results even without domain knowledge (for French) and without much fine-tuning. In future works, it would be certainly interesting to delve into a deeper study to ascertain the role of language and morphology in these models. Considering that tokenization is the most language-dependent step, this would involve testing several tokenizers, and pre-training the LMs from scratch on several monolingual corpora with adjusted language proportions, similarly to [7]. Clearly, this work would be very resource-intensive.

The similarity in results is not at all surprising, considering how close the three chosen languages are. Indeed, English has Germanic roots, while Italian and French are Romance languages (derived from Vulgar Latin), yet have developed very closely and have strong influences on each other. There are many differences that could be pointed out (gendered nouns and pronouns, liaisons, accents, etc.), but it is fair to consider them morphologically similar languages since they all belong to the fusional family.

Our results are suitable to prove the ease of application of pre-trained LMs and their convenience with respect to other traditional classification methods, as well as those based on LSTMs and MLPs with word embeddings. Despite our limited hyperparameter tuning imposed by a low-resource environment, these methods clearly show their value as one-and-for-all solutions for supervised TC. Moreover, the multilingual model XML-R was able to capture discriminative features in all three languages, in spite of the more limited per-language vocabulary.

## Future research challenges

The last few years have seen exciting developments in the field of Text Classification and NLP in general. Large-scale language models have achieved state-of-the-art results throughout NLP

literature, yet they are not infallible. These approaches face a set of challenges of their own regarding unexpected behaviors, true semantic understanding and harmful biases hidden in training data [3, 63]. Partially in response to these issues, new approaches are being researched, both to improve the reliability of LMs and to democratize their accessibility.

## Multitask learning

The domain and language dependence of language models is one notable issue faced by LMs. Ideally, these models should show general understanding of languages via pre-training on several modeling tasks. We have mentioned how one of the limiting factors for fair experimentation of recent NLP models in languages other than English is the lack of downstream, task-specific datasets. We have shown such scarcity in Italian, which we expect only to be worse in lower-resourced languages. In this regard, multitask learning is a novel approach to learning language embeddings through combining labeled data from multiple related tasks and fine-tuning simultaneously on all of them, thus producing cross-task embeddings. Liu et al [253] proposed multitask DNN that showed strong generalization capabilities on domains where little-to-no labeled data was provided. They also provide evidence that this strategy profits from a regularization effect that reduces overfitting on single tasks, thus making embeddings more universal. The XGLUE [243] and XTREME [246] frameworks extend this concept by introducing a standard evaluation procedure for cross-task and cross-lingual models. It's easy to see how these paradigms could help tackle common problems for NLP research, first and foremost the scarcity of labeled, task-specific data for various languages.

## Multilingual models

As shown, deep language models trained on large multilingual corpora are achieving excellent results [61, 70, 82], displaying a remarkable ability to extract semantic meaning across multiple languages. Given the computational complexity incurred by the development of Transformer-based models, it would perhaps be more desirable to push for the development of models that are able to generalize well to the widest array of languages possible. Multilingual models could help to prevent the newfound necessity of having to develop competitive monolingual language models for each language, which is becoming increasingly difficult due to the speed at which their dimension is growing. Furthermore, this would serve to benefit more under-represented languages, while partially addressing the justifiable ethical and environmental concerns related to the negative impact caused by the training process of these models [63]. Nonetheless, we have already addressed some of the limitations of these approaches, and how language-specific additions (such as a monolingual tokenizer) can improve performance.

New multilingual models are still being developed, often following the trend of what are colloquially defined as *large language models* (LLMs). Most of these projects are carried out by large tech companies, which are able to afford their vast computational costs; however, we point out notable scientific projects such as BigScience [254], currently being trained on 46 languages and more than 28 petaflops of textual data. It will be interesting to see the results of open projects such as these.

## Few-shot learning

Another direction being researched is that of *few-shot learning*, where models are shown very little to no labeled examples in the fine-tuning procedure. Therefore, the aim is the creation of generic models able to overcome the lack of task-specific datasets; an example is that of the aforementioned GPT-3 [69], which was one of the first works in recent years to display impressive results without large amounts of task-specific data or model parameter updating.

Other works have followed, training on even larger datasets from diverse sources and experimenting with sparsely activated modules to address the computational expensiveness of LLMs [255–258].

An interesting aspect of these models is that they have shown to have "strong" multilingual capabilities. Many of these models indeed include corpora in different languages in their pretraining; however, while their results are certainly impressive, they are still outshined by monolingual approaches on language-specific tasks.

### Reducing the size of language models

Though the trend of scaling larger and larger models still continues to this day, some developments are proposing smaller, generative LMs that have been shown to perform competitively when augmented with search/query information from retrieval databases [259, 260]. For instance, the developers of the Retrieval-Enhanced Transformer (RETRO) showcase performances on par with GPT-3, despite their model being 4% of the latter's size. As such, further research on the development of more reasonably sized models will be certainly a worthy endeavor.

## Conclusion

In this paper, we overview existing models for TC and study their applicability to other languages, utilizing Italian as our main point of perspective. Firstly, we discuss the relevancy of preprocessing operations as arguably the most language-specific steps in the TC pipeline. We introduce the most common tokenization techniques that are paired with the latest methods and we further expand on different approaches to project textual features in suitable feature spaces for machine processing. Deep neural language models are introduced, and, whenever appropriate, we comment on the challenges and possible solutions to their applicability to non-English languages, first and foremost their high-resource requirements. A brief overview of the last step in the pipeline, that of classification, is then given; state-of-the-art approaches are outlined, commenting on their different levels of language dependence. We then showcase a number of Italian TC datasets, a language we deem mid-resourced; to substantiate this claim, we also similarly search for French datasets. We make a comparison between the two as well as with equivalent English datasets, showing that both French and English have greater availability of large labeled corpora. Furthermore, we give new quantitative results on multilabel classification tasks in Italian, French, and English. In particular, we apply a few main representatives of the methods we described on News Classification and Topic Labeling, two subcategories of TC which are underrepresented outside of the English scope. Finally, we discuss future research challenges and directions of TC, with an emphasis on how they affect other languages.

## Supporting information

**S1 Datasets. Charts with dataset statistics.** Histograms with statistics about the RCV1/2 and Wiki datasets.
(PDF)

**S1 Appendix. Training and testing procedure.** Hyper-parameters used for training and relevant preprocessing operations.
(PDF)

## Acknowledgments

We want to thank the Academic Editor and the reviewers for their support and precious advice.

## Author Contributions

**Data curation:** Alessandro Zangari, Matteo Marcuzzo.

**Formal analysis:** Matteo Marcuzzo.

**Investigation:** Alessandro Zangari, Matteo Marcuzzo.

**Methodology:** Matteo Marcuzzo.

**Project administration:** Andrea Gasparetto.

**Software:** Alessandro Zangari, Matteo Marcuzzo.

**Supervision:** Andrea Gasparetto, Andrea Albarelli.

**Validation:** Andrea Gasparetto.

**Writing – original draft:** Andrea Gasparetto.

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
