## [Decision Letter · Decision Letter 0]

16 Mar 2022

PONE-D-22-01277A multilingual perspective on existing text classification methodsPLOS ONE

Dear Dr. Gasparetto,

Thank you for submitting your manuscript to PLOS ONE. After careful consideration, we feel that it has merit but does not fully meet PLOS ONE’s publication criteria as it currently stands. Therefore, we invite you to submit a revised version of the manuscript that addresses the points raised during the review process.

Please check the reviewers comments below.

We look forward to receiving your revised manuscript.

Kind regards,

Liviu-Adrian Cotfas

Academic Editor

PLOS ONE

Journal Requirements:

Reviewers' comments:

Reviewer's Responses to Questions

**Comments to the Author**

1. Is the manuscript technically sound, and do the data support the conclusions?

Reviewer #1: No

Reviewer #2: Yes

2. Has the statistical analysis been performed appropriately and rigorously? 

Reviewer #1: N/A

Reviewer #2: Yes

3. Have the authors made all data underlying the findings in their manuscript fully available?

Reviewer #1: No

Reviewer #2: Yes

4. Is the manuscript presented in an intelligible fashion and written in standard English?

Reviewer #1: Yes

Reviewer #2: Yes

5. Review Comments to the Author

Reviewer #1: The paper promises to present a "multilingual" perspective on existing text classification methods but then goes on to concentrate mostly on the languages Italian and French in comparison to English. Irrespective of the remainder of the paper, this seems like an arbitrary choice - given that there are thousands of languages and many dozen spoken at least by as many people as Italian or French, one has to wonder if the authors would propose to publish a similar paper on each random pair of two languages in the same fashion.

The paper provides quite a bit of useful information to anyone interested to get started with text classification (TC) on these two languages and also gives a quick overview over TC methods which can be useful to people getting started in that area.

However the paper does not really present any valuable scientific research questions worthy of getting published in a journal. They list their contributions as 1) "analysis" of TC procedures, focusing on text representation and how it affects the compatibility of thise models with lowe resourced languages` 2) showcase a variety of datasets for Italian TC, and similar for French, 3) proved quantitative results on "understudied" TC tasks.

Starting with 2) the paper does give a good overview of Italian/French and multilingual TC datasets, but while this is useful, there is not much scientific insight to be gained. No research questions are answered with regard to that contribution and there are now ample online web sites and search tools which will help to find that list of dataset with not too much effort. The authors mention for a number of different TC tasks, if there are fewer datasets for Italian or French than for English, but not much further information or analysis is presented.

The "analysis of TC procedures" is really a very shallow overview over the main TC methods and tasks which would be part of an introductory course on NLP. The overview does not really contain any additional information with regard to how specifically some of these methods may perform differently for different languages from a theoretical point of view or point at literature where such question would have been looked into. No discussion of how properties of languages like vocabulary size, morphology, word segmentation, script used, etc. could maybe impact different algorithms in different ways. Which is of course even harder with only two languages in focus, both of which are quite similar to English in these respects.

There is a whole section on "computational resources" without any indication how this relates to the topic of the paper.

For contribution 3) the authors apply a number of TC methods to similar datasets for topic labelling and news classification in the three languages and then compare the results. The authors point out some difficulties and inconsistencies with some of the datasets but do not analyse the impact of that on the results.

The distribution of classes is not reported, nor compared between languages. The choice of algorithms seems arbitrary and is not motivated by any relation to analysing the impact of language choice on the performance of the algorithm. Each method is run 4 times with different random data splits, but no other sources of randomness are mentioned (e.g. random initalization or optimization order for NN methods). The variance of the results is not reported, nor is there an indication of which differences between results could be considered significant.

Similar comparisons between different algorithms have been performed numerous times before with much more detailed analysis on what the reasons for the performance differences could be.

The comparison of results between languages is done only very superficially and there is some speculation of what could be possible causes, but there is no actual analysis or in depth-study into any of those potential causes, let alone into distinguishing between the impact of the language itself versus the impect of the specific corpus chosen for that language.

Overall I do not think the paper provides useful research insights that would justify publication in a journal. I can also not see a way for how the paper could get sufficiently improved by revising it.

Reviewer #2: The paper covers an interesting gap in our opinion, a good recent survey on the text classification problem. The fact that it is is a multilingual one, covering It, Fr and En is a plus.

Regarding the references, I think that you could add more results: for Reuters database, the works of fabrizio Sebastiani (eg Machine Learning in Automated Text Categorization), or the work of Rusu and Dinu (Rank Distance Aggregation as a Fixed Classifier Combining Rule for Text Categorization, CICLing 2010).

6. PLOS authors have the option to publish the peer review history of their article (what does this mean?). If published, this will include your full peer review and any attached files.

Reviewer #1: **Yes: **Johann Petrak

Reviewer #2: No

---

## [Author Response · Author response to Decision Letter 0]

17 May 2022

Dear PLOS ONE Editors and Reviewers,

We would like to thank the reviewers for their feedback. We have carefully considered your requests and suggestions, and addressed each and every one of them.

First off, in response to the editor's request, we have reviewed PLOS ONE's style requirements, and have done our best to update the manuscript to precisely meet them. 

In response to prompt (a) and (b), we note that the Reuters Corpora utilized in our experiments is owned by NIST, who acts as its sole distributor. In particular, the researchers are asked to sign an agreement by which 'the display, reproduction, transmission, distribution or publication of the information is prohibited' and 'summaries, analyses and interpretations of the linguistic properties of the information may be derived and published, provided it is not possible to reconstruct the information from these summaries'. These restrictions are documented at https://trec.nist.gov/data/reuters/reuters.html. On the other hand, we have made the extracted Wikipedia dumps available, together with information on how to use them. The code repository is available at https://gitlab.com/distration/dsi-nlp-publib, which contains a link to the cloud folder storing the datasets. 

We provide below a response to the received feedback and the explanation of all major changes introduced. To facilitate the revision process, we provide a copy of the manuscript annotated in red and blue color, respectively marking removed and added content.

We note that the manuscript has undergone significant changes; we believe such extensive additions were necessary to address the critical issues pointed out by the reviewers, reviewer 1 in particular. We now present the addressed issues point-by-point.

Reviewer 1 pointed out that presenting the paper as ``multilingual'' might be misleading, as it includes a small fraction of the overall thousands of languages in the world. As such, we have overhauled our work to put more emphasis on the fact that our aim was to gauge how well various classification methods could be adapted to a real-world scenario with limited resources, and that our perspective was mainly on the Italian language.

We also agree that the analysis of TC procedures was too shallow and not related enough to language-specific issues. In order to improve it, we have split it into three more organized sections. The Preprocessing section describes in more detail those preprocessing operations related to language, with a large focus on text segmentation. Indeed, this is the largest addition to the manuscript. The other two sections largely replace and expand the old 'TC procedures' section. The Text representation section goes into more detail about how text is projected into feature space as we underline the importance of proper text representation in any NLP procedure, especially from a linguistic point of view. Lastly, we've added a briefer section which provides some insight on the Classification step of the pipeline.

We've clarified the meaning of the 'computational resources' section as a severe bottleneck to experimentation with recent language models, which is not a linguistic issue per-se but it has direct impact on the possibility of evaluating these models. In fact, our limited resources were one of the reasons why we could not perform more accurate experimentation. We hope that, by improving the previous sections, as well as specifying this section in more detail, it is clearer that computational resource requirements bring daunting issues to the adoption of NLP methods in other languages.

Though we have slightly revised it, we find ourselves at a disagreement on the points brought about the analysis of datasets. Indeed, such research was meant to highlight the scarcity of downstream, task-specific datasets for the Italian language, which could only be proven by showing empirical data. Again, we disagree on the point that such datasets can be found with not too much effort, as can be proven by their scarce adoption in the literature. In our experience, many of these datasets are scattered around the web and cannot be reliably accessed through a single search tool. Moreover, we have found multiple references to existing datasets, only to find that they had been retired, made private or otherwise rendered unavailable. Nonetheless, we hope that this section can come across more smoothly after the revision done to the other parts of the manuscript.

We agree on the critiques related to the experimental section, and have added quite a few considerations to the analysis. We've added variance statistics in the performance results, as well as distribution of labels for all datasets. Details about initialization and optimization procedures were already present in the supplemental material provided, which has however been further expanded to explain the reason behind the choice of methods and to clarify further other technical details. 

Lastly, Reviewer 2 has asked us to add some references with regards to the analyzed Reuters database, which we have done.

We confirm that neither the manuscript nor any parts of its content are currently under consideration or published in another journal. All authors have approved the manuscript and agree with its submission to PLOS ONE.

Sincerely yours,

Andrea Gasparetto (corresponding author)

andrea.gasparetto@unive.it

---

## [Decision Letter · Decision Letter 1]

20 Jun 2022

A survey on text classification: practical perspectives on the Italian language

PONE-D-22-01277R1

Dear Dr. Gasparetto,

We’re pleased to inform you that your manuscript has been judged scientifically suitable for publication and will be formally accepted for publication once it meets all outstanding technical requirements.

Kind regards,

Sathishkumar V E

Academic Editor

PLOS ONE

Additional Editor Comments (optional):

Reviewers' comments:

Reviewer's Responses to Questions

**Comments to the Author**

1. If the authors have adequately addressed your comments raised in a previous round of review and you feel that this manuscript is now acceptable for publication, you may indicate that here to bypass the “Comments to the Author” section, enter your conflict of interest statement in the “Confidential to Editor” section, and submit your "Accept" recommendation.

Reviewer #3: (No Response)

2. Is the manuscript technically sound, and do the data support the conclusions?

Reviewer #3: (No Response)

3. Has the statistical analysis been performed appropriately and rigorously? 

Reviewer #3: (No Response)

4. Have the authors made all data underlying the findings in their manuscript fully available?

Reviewer #3: (No Response)

5. Is the manuscript presented in an intelligible fashion and written in standard English?

Reviewer #3: (No Response)

6. Review Comments to the Author

Reviewer #3: (No Response)

7. PLOS authors have the option to publish the peer review history of their article (what does this mean?). If published, this will include your full peer review and any attached files.

Reviewer #3: **Yes: **Usha Moorthy

---

## [Editor Report · Acceptance letter]

24 Jun 2022

PONE-D-22-01277R1 

A survey on text classification: practical perspectives on the Italian language 

Dear Dr. Gasparetto:

I'm pleased to inform you that your manuscript has been deemed suitable for publication in PLOS ONE. Congratulations! Your manuscript is now with our production department. 

Kind regards, 

on behalf of

Dr. Sathishkumar V E 

Academic Editor

PLOS ONE